# Thermal traits for reproduction and recruitment differ between Arctic and Atlantic kelp *Laminaria digitata*

**Neusa Martins**[1,2]*, **Gareth A. Pearson**[1], **Julien Bernard**[1], **Ester A. Serrão**[1], **Inka Bartsch**[2]

**1** Centre of Marine Sciences (CCMAR), University of Algarve, Faro, Portugal, **2** Alfred-Wegener-Institute, Helmholtz Center for Polar and Marine Research, Bremerhaven, Germany

* nemartins@ualg.pt

**Data Availability Statement:** All relevant data are within the manuscript and its Supporting Information files.

## Abstract

The plasticity of different kelp populations to heat stress has seldom been investigated excluding environmental effects due to thermal histories, by raising a generation under common garden conditions. Comparisons of populations in the absence of environmental effects allow unbiased quantification of the meta-population adaptive potential and resolution of population-specific differentiation. Following this approach, we tested the hypothesis that genetically distinct arctic and temperate kelp exhibit different thermal phenotypes, by comparing the capacity of their microscopic life stages to recover from elevated temperatures. Gametophytes of *Laminaria digitata* (Arctic and North Sea) grown at 15˚C for 3 years were subjected to common garden conditions with static or dynamic (i.e., gradual) thermal treatments ranging between 15 and 25˚C and also to darkness. Gametophyte growth and survival during thermal stress conditions, and subsequent sporophyte recruitment at two recovery temperatures (5 and 15˚C), were investigated. Population-specific responses were apparent; North Sea gametophytes exhibited higher growth rates and greater sporophyte recruitment than those from the Arctic when recovering from high temperatures, revealing differential thermal adaptation. All gametophytes performed poorly after recovery from a static 8-day exposure at 22.5˚C compared to the response under a dynamic thermal treatment with a peak temperature of 25˚C, demonstrating the importance of gradual warming and/or acclimation time in modifying thermal limits. Recovery temperature markedly affected the capacity of gametophytes to reproduce following high temperatures, regardless of the population. Recovery at 5˚C resulted in higher sporophyte production following a 15˚C and 20˚C static exposure, whereas recovery at 15˚C was better for gametophyte exposures to static 22.5˚C or dynamic heat stress to 25˚C. The subtle performance differences between populations originating from sites with contrasting local *in situ* temperatures support our hypothesis that their thermal plasticity has diverged over evolutionary time scales.

**Funding:** This work was supported by a Pew Marine Fellowship (to EAS) and the Foundation for Science and Technology (FCT) of Portugal through PTDC/MAR-EST/6053/2014, UID/Multi/04326/2019, BIODIVERSA/0004/2015, SFRH/BPD/122567/2016 to NM (in transitional norm DL 57/2016/CP1361/CT0039) and SFRH/BSAB/150485/2019.

**Competing interests:** The authors have declared that no competing interests exist.

**Abbreviations:** DHS, dynamic heat stress; PES, Provasoli enriched seawater; RGR, relative growth rate.

## Introduction

Thermal adaptation studies have recently focused widely on anthropogenically induced climate change in terms of the consequences of shifts in mean temperatures [1, 2]. However, adaptive changes among populations may be driven be the extreme thermal values experienced, such as the frequency, intensity and persistence of extreme climatic events [3]. Such complex conditions of environmental change, including the rate and duration of heat spikes, might have important implications for ecophysiological performance [4] and consequent population divergence, but remain insufficiently understood.

Although many studies have investigated thermal responses of marine species, most do not consider intraspecific variation in adaptive traits between populations along distributional ranges [5, 6]. Species with broad geographical distributions, inhabiting a diversity of local thermal regimes, may diverge with distinct genetic and phenotypic features over evolutionary time scales [6]. Marginal habitats may contain populations with unique genetic diversity and corresponding phenotypic characteristics resulting in important persistence capability and conservation value under the impact of global change [7–11]. Distinguishing genetic and phenotypic variation among populations within a species is rare, as intraspecific acclimation versus adaptation along distributional ranges are difficult to disentangle, requiring common garden conditions to verify if a generation of individuals that develop from meiospores/zygotes in the same conditions still retained different ecophysiological responses to the same thermal extremes.

Studies that compare the variability of thermal responses of marine organisms across distinct populations have followed various approaches to answer distinct questions; using the same thermal stress under common garden conditions can show whether populations of the same species have adaptive differences to thermal tolerance limits, non-common garden conditions address the question of phenotypic differentiation which can be due to acclimation or adaptation, while delta approaches (where temperature changes of a common magnitude delta relative to those prevailing at the population site are applied) address the question of direct local ecological effects under global warming scenarios acting locally. A distinct issue is the rate of change: experimentally-determined thermal tolerances of species are highly dependent on the chosen methodology and experimental conditions, namely whether temperature stress is applied in a static or dynamic (gradual) way [12]. Experiments with a dynamic stress increase rate better simulate ecologically relevant *in situ* conditions, incorporating more naturalistic shifts in temperature as opposed to the stable stress level applied in static methods [12]. Available data comparing both methods is still limited, particularly for species that occur in cold-temperate regions, that might be particularly sensitive to selective thermal stress levels, such as the marine forests of macroalgae commonly named kelp.

Kelp is a common name used for large brown bio-engineering algae, of several Phaeophycean orders (mostly Laminariales, but also used for some Tilopteridales, Fucales and Desmarestiales). Kelps form marine forests along rocky polar to temperate coastlines worldwide [13], even including very deep reefs in tropical regions. Kelp forests provide a wide range of ecosystem goods and services, both directly as source of food, alginates and pharmaceutical products and indirectly by forming biogenic and structural habitats for numerous ecologically and economically important marine species [13, 14]. Many populations of these habitat-forming algae are currently under threat from global warming, with large-scale declines in their abundance and range shifts occurring worldwide [15–17]. This has especially been observed in warming hotspots or near distributional equatorward edges (e.g. *Laminaria digitata* in France [18]; *Saccharina latissima* in Norway [17]; *Laminaria hyperborea*, *Laminaria ochroleuca* and *Saccorhiza polyschides* in Spain [19] and Portugal [20, 21]; *Ecklonia radiata* in Australia [15, 22]).

The perennial kelp species, *Laminaria digitata* (Hudson) J.V. Lamouroux occurs in the lower intertidal and shallow sublittoral of rocky habitats across large areas of North Atlantic to Arctic coastlines, forming dense marine forests and functioning as foundation species in these ecosystems. In the NE Atlantic, the distribution of *L. digitata* ranges from the Arctic (Svalbard) to southern Brittany, France [23]. As in typical kelp species, *L. digitata* exhibits a heteromorphic haplodiplontic life cycle with alternation of microscopic stages (meiospores, gametophytes and microscopic sporophytes) and macroscopic sporophytes [23, 24]. Mature sporophytes produce meiospores that develop into haploid female or male gametophytes, which under favourable conditions develop sperm and eggs that fertilize to regenerate diploid sporophytes [25]. The various ontogenetic stages of complex life cycles possess different thermal response pattern and survival limits. In *L. digitata*, sporogenesis and gametogenesis require a lower temperature window than sporophyte and gametophyte growth and survival [26–29], thus an increase in temperature may delay reproductive development. Haploid and diploid phases also vary; gametophytes tolerate higher temperatures for growth and survival than sporophytes [27, 30]. Vegetative gametophytes have been suggested to play a role analogous to plant seeds [30, 31], postponing fertility until favourable environmental conditions prevail. Therefore, these stages are especially important for kelp recruitment and may ensure species survival in populations that experience massive sporophyte mortality due to extreme environmental events [32–34].

Common garden experiments as well as knowledge about trait plasticity in different kelp populations are scarce. The aim of this study was to detect adaptive differentiation between populations in their response to extreme thermal conditions. To achieve this goal, we compare two populations of *L. digitata* originating from locations with contrasting thermal histories (Arctic; Spitsbergen vs North Sea; Helgoland), that had been kept under the same stable long-term laboratory conditions for the whole life since meiospores germinated, as required to infer adaptation rather than acclimation. Thus, differences should theoretically show evolutionary adaptations consistent with the thermal history at the respective locations of the source populations compared. As pre-investigations showed that thermal growth optima of different geographical isolates seem to be quite stable [27, 35], we assumed that sub-lethal to lethal conditions might better reveal population differences. We therefore experimentally investigated the reproductive performance of microscopic gametophytes and subsequent sporophyte recruitment capacity after extreme thermal conditions and known optimal controls. The major question was whether there is genetically based phenotypic differentiation among individuals from distinct populations when exposed to the species lethal and sub-lethal limits. The question of this study was not to compare population responses to local levels of *in situ* global warming conditions.

## Materials and methods

### Algal material

Two populations of *L. digitata* from distinct environmental conditions were used. Five mature sporophytes were collected during low tide from Helgoland, North Sea, Germany (54˚10'39"N, 7˚53'36"E) in September 2015 and by divers in Kongsfjorden, Spitsbergen, Svalbard (78˚59'06.1"N, 11˚57'47.6"E) in June 2015. In Kongsfjorden the summer sea-surface temperature has been recently recorded to be around 7–8˚C [36], while in Helgoland the summer sea-surface temperature regularly reaches 18˚C and is often higher [37]. Helgoland represents a boundary site for survival of *L. digitata* [28], as summer seawater temperatures are comparable to or even higher than those at the southern distribution limit in Brittany [28, 38].

Sori were cleaned and meiospores from each individual were released separately into sterile seawater. After the development of gametophytes, female and male gametophyte stock cultures (AWI culture numbers—Helgoland: 3435, 3436, 3439–3444, 3447, 3448; Spitsbergen: 3467–3476) from each individual were established separately in Petri dishes according to the protocol of Bartsch [39]. Vegetative cultures were maintained at 15˚C under 3 μmol photons m$^{-2}$ s$^{-1}$ of red light (LED Mitras daylight 150 controlled by ProfiLux 3, GHL Advanced Technology, Kaiserslautern, Germany), 16:8 h light:dark (LD) cycle in sterile full strength Provasoli enriched seawater (PES; Provasoli [40], modifications: HEPES buffer instead of TRIS, double concentration of Na2glycerophosphate) until the start of the experiment (i.e., ca. 3 y). The seawater of these stock cultures was changed monthly.

## Experimental setup

The same amount of vegetative female and male gametophytes derived from the five *L. digitata* individuals from each population were mixed separately for each sex and gently ground with pestle and mortar into fragments averaging a few cells each. The suspensions were sieved and diluted in sterile seawater to produce 4 stock solutions (Helgoland ♀; Helgoland ♂; Spitsbergen ♀; Spitsbergen ♂) of gametophytes with lengths between 50 to 100 μm. Each stock solution was a mix of 5 strains of female or male gametophytes. Densities from each single-sex stock were calculated. Female and male gametophyte stock solutions from each population were then combined and the volume needed to achieve a gametophyte density of ~400 gametophytes cm$^{-2}$ was added to Petri dishes (6.3 cm diameter, height 6.4 cm) containing 4 thick cover slips (n˚ 3) each and 100 ml of 10% PES. Four replicate Petri dishes were used for each treatment (2 populations × 6 temperature/dark treatments × 4 replicates = 48 Petri dishes in total) each containing an equal mixture of female and male gametophytes.

The gametophytes were allowed to settle and establish at 15˚C under 3 μmol photons m$^{-2}$ s$^{-1}$ of red light for 5 days. After this initial period the gametophytes were transferred to each target static temperature treatment (control at 15˚C, heat stress at 20˚C, 22.5˚C and 25˚C ± 0.1˚C) and to a dynamic temperature treatment for 8 experimental days. In the dynamic heat stress (25˚C DHS) the temperature was slowly increased from 15˚C to 25˚C at a warming rate of 2–3˚C day$^{-1}$ and the temperature of 25˚C was kept over a period of 2 days. Then, the seawater temperature was decreased from 25˚C back to 15˚C (again with a cooling rate of 2–3˚C day$^{-1}$). In total, the dynamic heat exposure also lasted 8 days and consisted of 3 days of warming, 2 days at peak temperature (25˚C) and 3 days of cooling (see Fig 1 for detailed experimental design). Four additional Petri dishes were prepared per population and maintained at 15˚C in darkness (15˚C dark), to simulate microscopic gametophytes growing in the very shaded subcanopy under parental sporophytes or under sediment (e.g., [41, 42]). The temperature of 15˚C was used as control because it is the optimal temperature for the growth and reproduction of *L. digitata* gametophytes [26, 27, 29]. On the other hand, the sub-lethal and lethal seawater temperatures of 20˚C, 22.5˚C and 25˚C for *L. digitata* gametophytes [30] were chosen to easily check for functional differences between populations. Experiments were conducted in temperature controlled water-baths (Huber Variostat with Pilot ONE, Offenburg, Germany), irradiance was set to 15 μmol photons m$^{-2}$ s$^{-1}$ (white LED light, in order to induce gametogenesis) measured with a LI-COR LI-185B Photometer (LI-COR-Biosciences, Lincoln, NE, USA) under a photoperiod of 16:8h light:dark. The 16:8h light:dark regime was used for both populations as it is a good approximation for summer daylengths along the whole distributional range of the species.

After the thermal and the dark treatments two cover slips with gametophytes from each replicate were transferred to a new Petri dish (5.3 cm diameter, height 1.5 cm), filled with 12

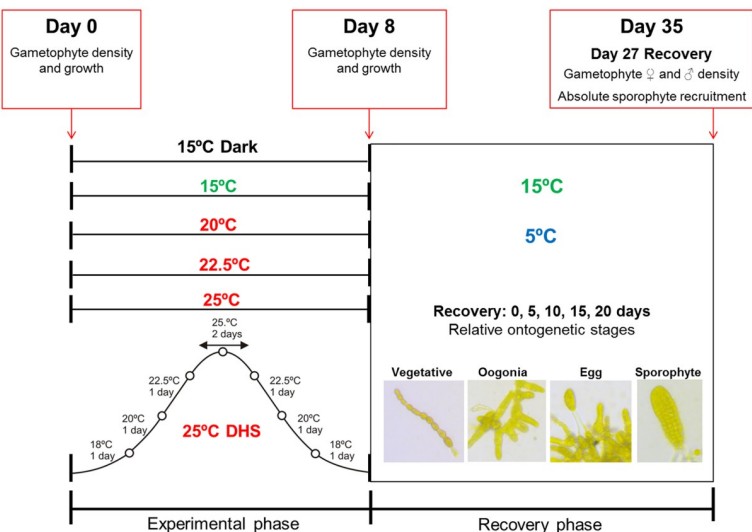

**Fig 1. Experimental design.** The diagram shows the different thermal/dark treatments (15°C dark, 15°C, 20°C, 22.5°C, 25°C and a dynamic heat stress with a peak temperature of 25°C: 25°C DHS) applied for 8 days in the gametophytes of *Laminaria digitata* from two populations and the subsequent recovery phase at 5°C and 15°C for 27 days. Gametophyte density and growth, development of ontogenetic stages and sporophyte recruitment parameters evaluated at different time points are also shown in the diagram.

ml of 10% PES, and exposed to 5°C and the other two to 15°C for 27 days of recovery. Culture medium was changed every week by the replacement of 6 ml of 10% PES per Petri dish.

## Gametophyte growth, ontogenetic stages of gametogenesis and sporophyte recruitment

**Gametophyte density.** To assess whether the treatments affected the survival of the gametophytes, the combined density of female and male gametophytes was determined at the beginning (day 0) and at the end (day 8) of all treatments, since it was difficult to clearly differentiate female and male gametophytes during this stage of development. At the end of the recovery phase (day 35 = 8 days of treatments + 27 days of recovery) the density of female and male gametophytes was evaluated separately. A minimum of 300 multicellular gametophytes was counted per replicate at each sampling point.

**Gametophyte growth.** Gametophyte area was quantified on day 0 and day 8 of each treatment by processing photographic data obtained from an Olympus CKX41 inverted microscope (Olympus Co., Tokyo, Japan) with a Zeiss Axiocam ERC5s microscope camera (Zeiss, Jena, Germany), using ImageJ software [43]. Ten fields of view (100× magnification) per replicate were randomly photographed, and the area of all gametophytes (female and male) present in each field of view was measured. The gametophyte average area was determined per replicate. If a female gametophyte became fertile and formed oogonia, the area of the oogonia was included. In contrast, if eggs and developing sporophytes were detected they were not included in the area measurements.

Relative growth rates (RGR) were estimated using the following formula:

Relative growth rate $(\text{day}^{-1}) = [\ln(\text{final area}) - \ln(\text{initial area})]/T$, where T is the culture period (days).

**Quantification of ontogenetic stages.** The relative occurrence of four ontogenetic stages of female gametophytes (vegetative state, gametophytes with oogonia, gametophytes with eggs released and gametophytes with sporophytes attached) was quantified at the end of the thermal/dark treatments (day 8 of treatment = day 0 of recovery) and every 5 days for 20 days after recovery in $\geq$ 300 female gametophytes per replicate using an Olympus CKX41 inverted microscope. For each female gametophyte the most advanced developmental stage was recorded. Gametophytes were considered to be in the oogonia, egg release or sporophyte stage if at least 1 cell per multicellular gametophyte had entered this developmental stage. Juvenile sporophytes were differentiated from released eggs if a first cell division was visible.

**Recruitment of juvenile sporophytes.** Recruitment capacity of juvenile sporophytes was evaluated through the relative presence of female multicellular gametophytes with sporophytes after 20 days of recovery and the absolute number of sporophytes per $cm^2$ after 27 days of recovery. The absolute number of sporophytes was evaluated using an Olympus CKX41 inverted microscope and a total of 50 fields of view (100 X magnification) were analysed per replicate.

Significant differences in the absolute number of sporophytes were observed between the two populations at 15°C control (S1 Fig). The Arctic population produced 950–2000 sporophytes $cm^{-2}$ compared with 450–800 sporophytes $cm^{-2}$ in the North Sea population. As these may be a result of differences in the initial number of cells per female multicellular vegetative gametophyte between populations, this parameter was normalized to the proportion of the control treatment (15°C) at the recovery temperature of 5°C.

## Statistics

Data were analysed with the PERMANOVA module within Primer 6 software [44, 45]. The gametophyte density after 8 days and the relative growth rate data were evaluated under a two-factor design, with treatment and population as fixed factors, whereas the normalized absolute sporophyte density and the female and male gametophyte density after 27 days of recovery were analysed under a three-factor design, with treatment, population and recovery temperature as fixed factors. As all the *L. digitata* gametophytes from both populations died after 8 days at 25°C, this treatment was excluded from the analysis. PERMDISP tests were performed to test the homogeneity of multivariate dispersions. The normalized sporophyte density was transformed via Box-Cox transformation ($\lambda$ = 0.51) as dispersion tests were significant. Post-hoc pair-wise t-test comparisons were performed to identify differences between treatments whenever a significant main effect or interaction was found. Univariate analyses were performed with Euclidian distances and 9999 permutations. Differences were considered significant at $p < 0.05$.

## Results

### Effect of heat stress and darkness: Gametophyte density and survival

Mean initial gametophyte density was 395 gametophytes $cm^{-2}$ and did not significantly vary between treatments or populations (Table 1). After 8 days, no gametophytes survived under the highest continuous temperature, 25°C (Fig 2). Excluding this treatment from the analysis, no significant interactions or main effects of treatments or populations were observed (Table 1; Fig 2), thus overall gametophyte survival (not separated by sex) was the same in all these conditions.

**Table 1. PERMANOVA for the effects of population and treatments on the gametophyte density of *Laminaria digitata*.**

| Factor | df | SS | MS | Pseudo-F | P(perm) |
|---|---|---|---|---|---|
| Density at day 0 | | | | | |
| Population | 1 | 588.58 | 588.58 | 0.47 | 0.495 |
| Treatment | 5 | 2647.90 | 529.57 | 0.42 | 0.830 |
| Population × Treatment | 5 | 284.92 | 56.98 | 0.04 | 0.998 |
| Residual | 36 | 45138 | 1253.80 | | |
| Density at day 8 | | | | | |
| Population | 1 | 133.11 | 133.11 | 0.14 | 0.715 |
| Treatment | 4 | 4415.20 | 1103.80 | 1.15 | 0.357 |
| Population × Treatment | 4 | 6187 | 1546.80 | 1.61 | 0.194 |
| Residual | 30 | 28808 | 960.27 | | |

df, degrees of freedom; SS, sum of squares; MS, mean sum of squares; Pseudo-F, F value by permutation.

### Gametophyte growth

Relative growth rates (RGR) of gametophytes showed significant population × treatment interactions (Fig 3; Table 2). North Sea gametophytes showed higher RGR at 15°C (1.6-fold) and 20°C (1.3-fold) and at the dynamic heat stress (25°C DHS) treatment (2.4-fold) compared to Arctic gametophytes. The RGR of Arctic gametophytes was 3.2-fold lower at 22.5°C, 25°C

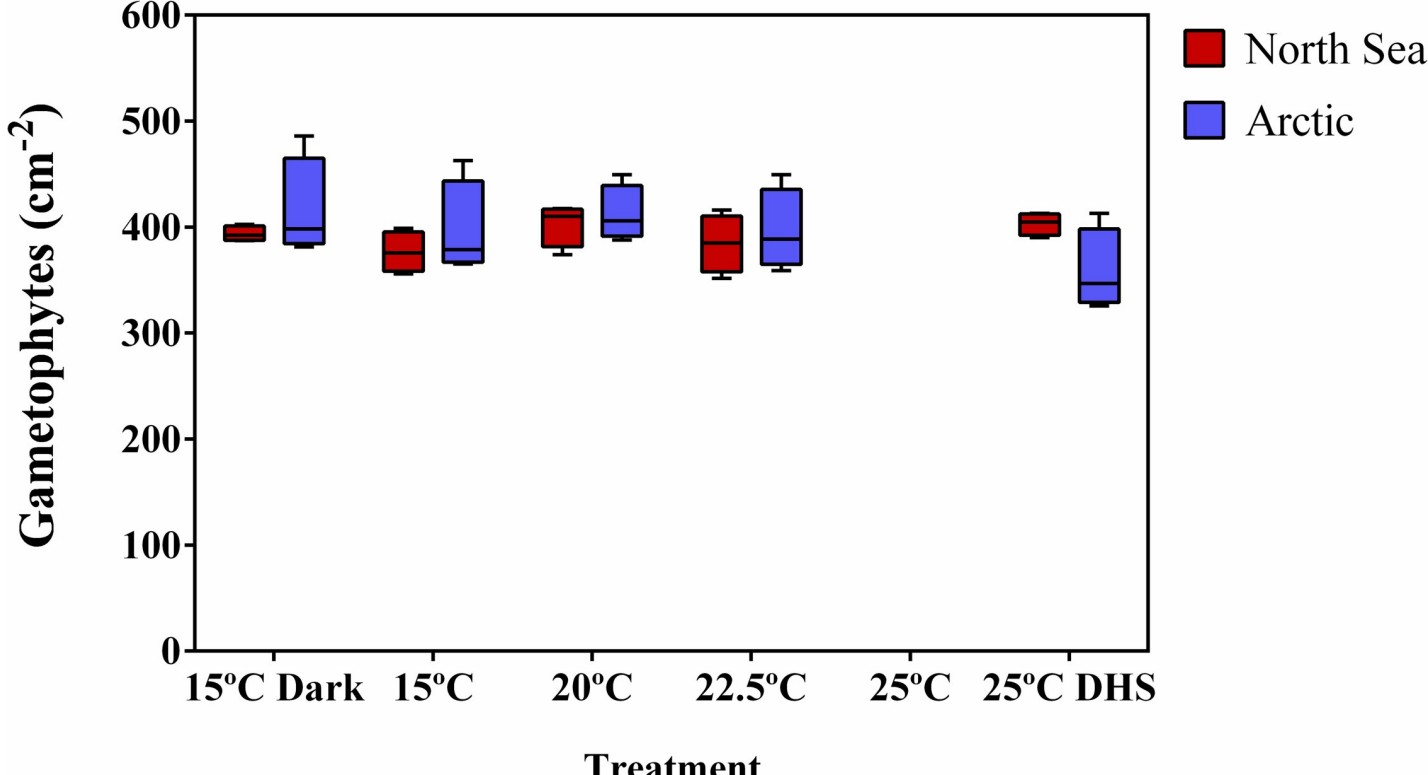

**Fig 2. Effect of heat stress and darkness on gametophyte density.** Gametophyte density from the North Sea and Arctic populations of *Laminaria digitata* after 8 days in different treatments (15°C dark, 15°C, 20°C, 22.5°C, 25°C and 25°C DHS). Box plots with median, boxes for 25th and 75th percentiles and whiskers indicating min and max values (n = 4). No significant differences between treatments or populations were detected (excluding the 25°C treatment). See Table 1 for statistics.

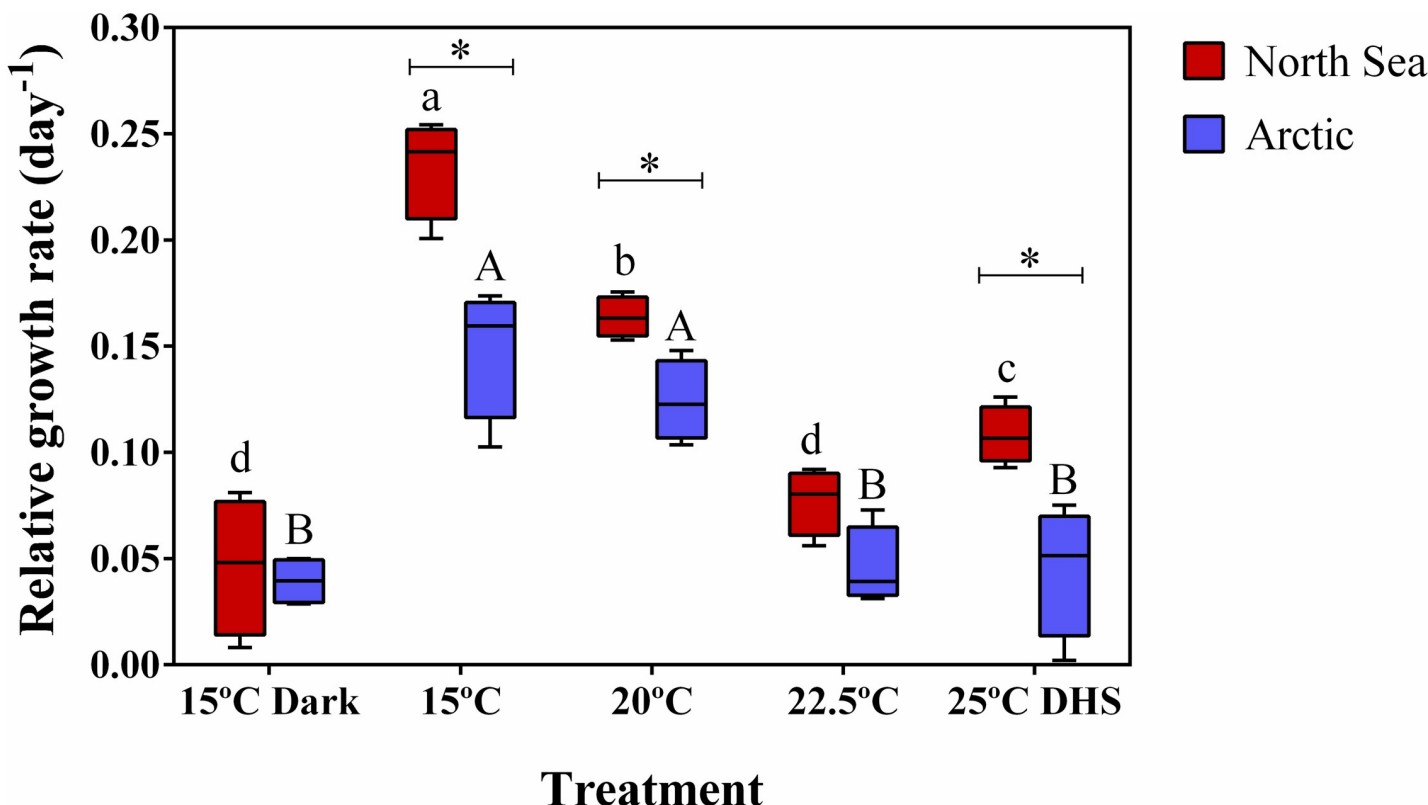

**Fig 3. Effect of heat stress and darkness on gametophyte growth.** Relative growth rate of gametophytes from the North Sea and Arctic populations of *Laminaria digitata* after 8 days in different treatments (15˚C dark, 15˚C, 20˚C, 22.5˚C and 25˚C DHS). Box plots with median, boxes for 25th and 75th percentiles and whiskers indicating min and max values (n = 4). * indicates a significant difference between populations per treatment (p<0.05). For each population, different letters above boxplot bars (lowercase letters for the North Sea population and upper case letters for the Arctic population) indicate differences between treatments (p<0.05). See Table 2 for statistics.

DHS and in darkness compared to 15˚C and 20˚C (Fig 3). In contrast, the RGR of North Sea gametophytes was reduced by all the warmer temperature treatments (1.4-fold reduced at 20˚C, 2.2-fold at 25˚C DHS and 3.0-fold at 22.5˚C) and even more by the dark treatment (5.1-fold) compared to the temperature control at 15˚C.

### Recovery capacity of gametophytes after heat stress: Time course of ontogeny

Ontogeny after heat stress during the recovery phase at 5˚C and 15˚C is shown in Fig 4. At day 0 of the recovery period (after 8 days of temperature exposure), only vegetative gametophytes

**Table 2. PERMANOVA for the effects of population and treatments on the relative growth rate for gametophyte area of *Laminaria digitata* after 8 days.** The post-hoc results are presented in Fig 3.

| Factor | df | SS | MS | Pseudo-F | P(perm) |
|---|---|---|---|---|---|
| Population | 1 | 205.66 | 205.66 | 41.93 | **<0.001** |
| Treatment | 4 | 1247.40 | 311.84 | 63.59 | **<0.001** |
| Population × Treatment | 4 | 72.82 | 18.20 | 3.71 | **0.013** |
| Residual | 30 | 147.13 | 4.90 | | |

Significant interactions or main effects are highlighted in bold. df, degrees of freedom; SS, sum of squares; MS, mean sum of squares; Pseudo-F, F value by permutation.

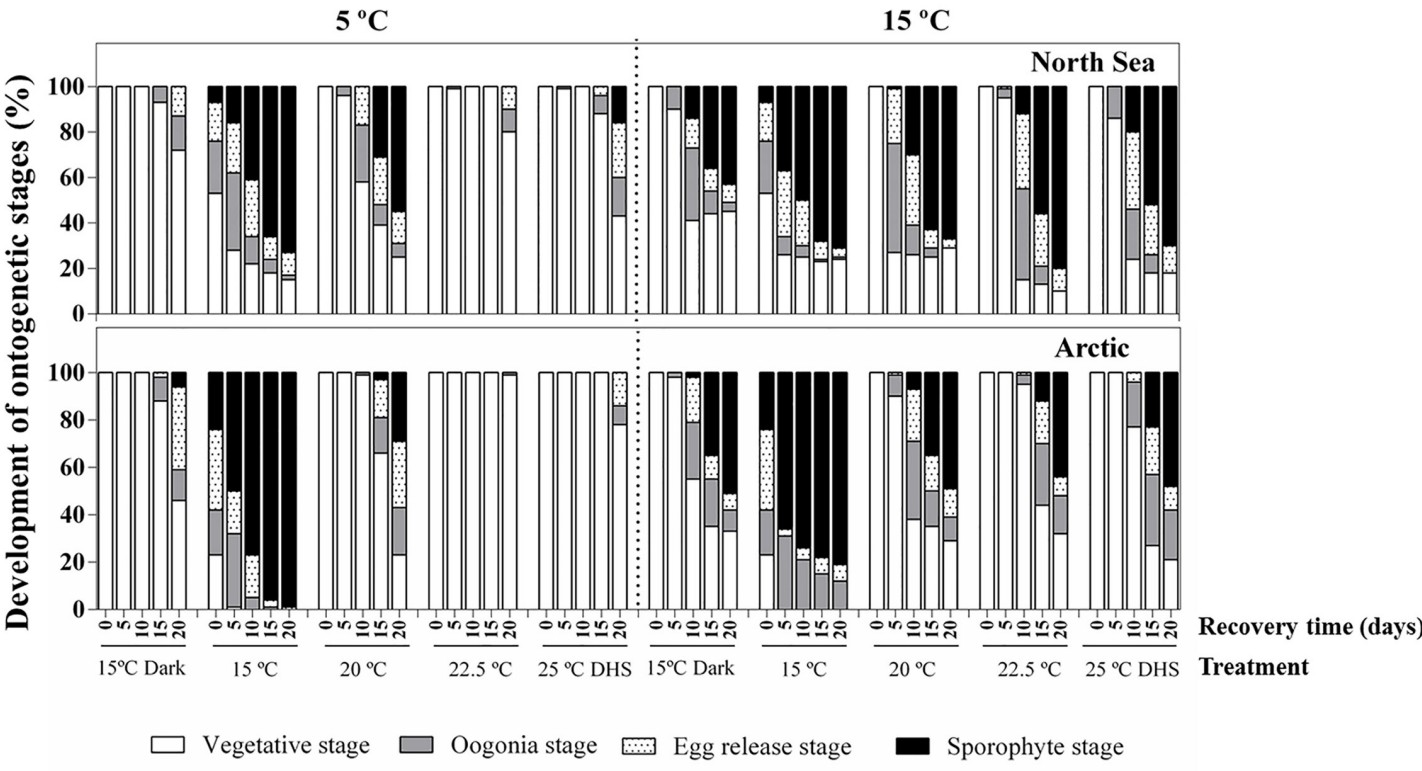

**Fig 4. Development of gametogenesis stages over recovery.** Development of ontogenetic stages in the North Sea and Arctic populations of *Laminaria digitata* over recovery time at 5˚C and 15˚C from different treatments (15˚C dark, 15˚C, 20˚C, 22.5˚C and 25˚C DHS) (mean values, n = 4). Counting was performed every 5 days over a period of 20 days. SE-values are omitted for clarity. Note that day 0 of the recovery phase is day 8 of the temperature exposure phase.

were observed at 20˚C, 22.5˚C, 25˚C DHS and in darkness in both populations. In contrast, at the control temperature of 15˚C a considerable proportion of female gametophytes from both populations had already become fertile at the same time point (Arctic: 19% oogonia, 34% eggs, 24% sporophytes; North Sea: 23% oogonia, 17% eggs, 7% sporophytes).

In both populations, gametophytes pre-exposed to 22.5˚C had the slowest development of fertility (oogonia, eggs and sporophytes production) at both recovery temperatures. Lower proportions of female gametophytes became fertile at the lower recovery temperature of 5˚C than at 15˚C in all treatments except 15˚C and in both populations (Fig 4).

Arctic gametophytes pre-exposed to 15˚C, exhibited a more rapid and complete gameto-genesis during recovery at 5˚C or 15˚C compared to North Sea gametophytes (15–25% of the gametophytes did not become fertile even after 20 days of recovery). In contrast, when pre-exposed to 20˚C, 22.5˚C and 25˚C DHS, Arctic gametophytes showed slower gametogenesis at both recovery temperatures compared to North Sea gametophytes.

### Recruitment capacity of juvenile sporophytes

The normalized sporophyte density after 27 days showed significant population × treatment and population × recovery temperature interactions (Fig 5A and 5B; Table 3). Recruitment success was higher in North Sea gametophytes recovering from 20˚C, 22.5˚C and 25˚C DHS exposure than in Arctic gametophytes (Fig 5A and 5B). Gametophytes from the 15˚C control exhibited the highest densities of sporophytes in both populations followed by the 20˚C pre-treatment. In the Arctic population, the gametophytes pre-exposed to 25˚C DHS had lower

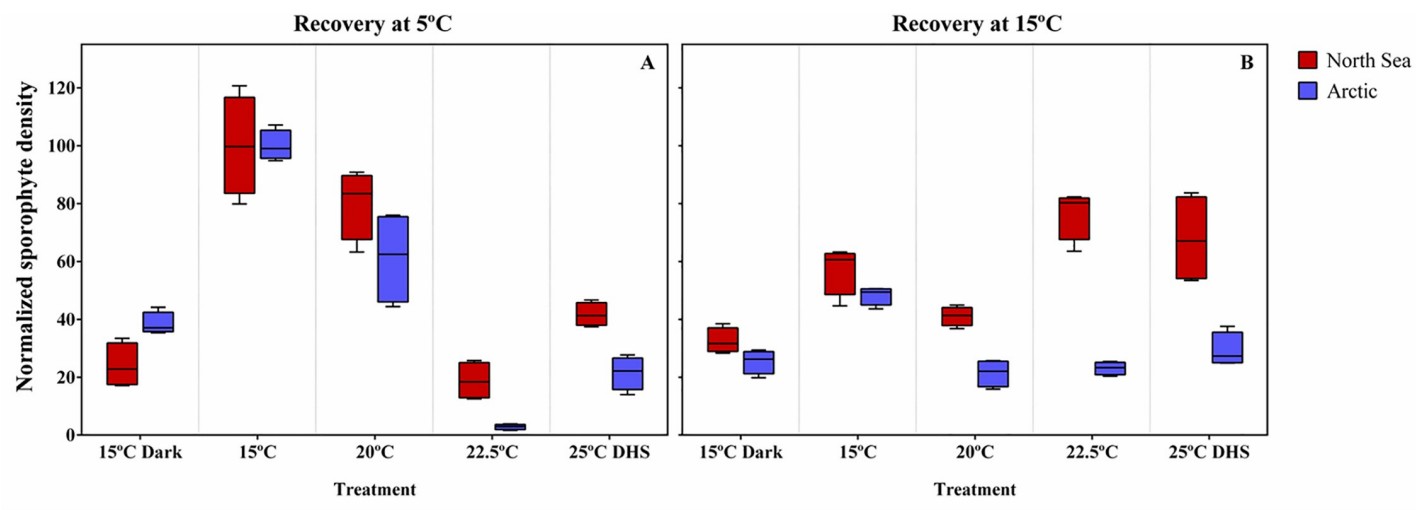

**Fig 5. Recruitment capacity of juvenile sporophytes after recovery.** Absolute number of sporophytes from the North Sea and Arctic populations of *Laminaria digitata* after 27 days of recovery at 5˚C (A) and 15˚C (B) from different treatments (15˚C dark, 15˚C, 20˚C, 22.5˚C and 25˚C DHS). Please note that density values were normalized to the control treatment value for each population (adjusted mean = 100%). Box plots with median, boxes for 25th and 75th percentiles and whiskers indicating min and max values (n = 4). See Table 3 for statistics.

sporophyte densities compared to 15˚C, but the lowest densities were observed in the gametophytes pre-exposed to 22.5˚C. Similarly, in the North Sea gametophytes the pre-exposure to 25˚C DHS and 22.5˚C decreased sporophyte density 1.4-fold and 1.7-fold, respectively compared to 15˚C, while the darkness pre-treatment further decreased sporophyte density 2.8-fold.

North Sea gametophytes showed higher sporophyte densities compared to Arctic gametophytes at both recovery temperatures (Fig 5A and 5B). Recovery at 5˚C increased the sporophyte density in the Arctic gametophytes compared to a recovery at 15˚C, while sporophyte densities did not significantly vary between recovery temperatures in the North Sea population.

The normalized sporophyte density also differed significantly due to the interaction recovery temperature × treatment (Fig 5A and 5B; Table 3). Gametophytes pre-exposed to 15˚C control and 20˚C recovered better at 5˚C exhibiting significantly higher sporophyte

**Table 3. PERMANOVA for the effects of population, recovery temperature and treatments on the recruitment capacity of juvenile sporophytes of *Laminaria digitata*.**

| Factor | df | SS | MS | Pseudo-F | P(perm) |
|---|---|---|---|---|---|
| Population | 1 | 169.74 | 169.74 | 116.64 | **<0.001** |
| Temperature | 1 | 1.84 | 1.84 | 1.27 | 0.267 |
| Treatment | 4 | 585.28 | 146.32 | 100.55 | **<0.001**[*] |
| Population × Temperature | 1 | 26.85 | 26.85 | 18.45 | **<0.001**[*] |
| Population × Treatment | 4 | 147.42 | 36.86 | 25.33 | **<0.001**[*] |
| Temperature × Treatment | 4 | 542.79 | 135.70 | 93.24 | **<0.001**[*] |
| Population × Temperature × Treatment | 4 | 5.69 | 1.43 | 0.98 | 0.420 |
| Residual | 60 | 87.32 | 1.46 | | |

Significant interactions or main effects are highlighted in bold.

[*] PERMDISP, p<0.05. df, degrees of freedom; SS, sum of squares; MS, mean sum of squares; Pseudo-F, F value by permutation.

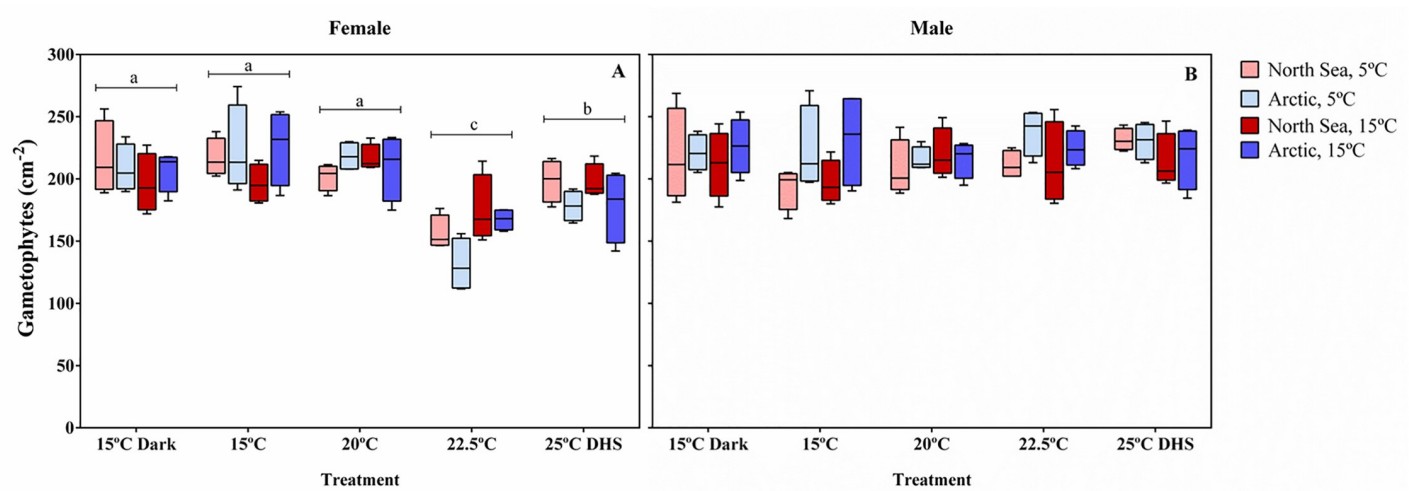

**Fig 6. Female and male gametophyte survival after recovery.** Female (A) and male (B) gametophyte densities from the North Sea and Arctic populations of *Laminaria digitata* after 27 days of recovery at 5˚C and 15˚C from different treatments (15˚C dark, 15˚C, 20˚C, 22.5˚C and 25˚C DHS). Box plots with median, boxes for 25th and 75th percentiles and whiskers indicating min and max values (n = 4). Different letters indicate significant differences among means for each treatment. See Table 4 for statistics.

recruitment than at 15˚C. In contrast, recovery at 15˚C enhanced sporophyte recruitment compared to a recovery at 5˚C in gametophytes previously exposed to 22.5˚C and 25˚C DHS conditions (Fig 5A and 5B). At the 5˚C recovery temperature gametophytes from the control had higher sporophyte densities compared to all the other treatments, followed by the 20˚C

**Table 4. PERMANOVA for the effects of population, recovery temperature and treatments on the female and male gametophyte density of *Laminaria digitata* after 27 days of recovery.** The post-hoc results are presented in Fig 6.

| Factor | df | SS | MS | Pseudo-F | P(perm) |
|---|---|---|---|---|---|
| Female gametophyte density | | | | | |
| Population | 1 | 109.35 | 109.35 | 0.25 | 0.617 |
| Temperature | 1 | 100.19 | 100.19 | 0.23 | 0.627 |
| Treatment | 4 | 36644 | 9160.90 | 20.98 | **<0.001** |
| Population × Temperature | 1 | 280.44 | 280.44 | 0.64 | 0.419 |
| Population × Treatment | 4 | 3941 | 985.25 | 2.26 | 0.069 |
| Temperature × Treatment | 4 | 3730 | 932.50 | 2.14 | 0.083 |
| Population × Temperature × Treatment | 4 | 1475.80 | 368.96 | 0.84 | 0.510 |
| Residual | 60 | 26202 | 436.7 | | |
| Male gametophyte density | | | | | |
| Population | 1 | 3256.80 | 3256.80 | 6.20 | **0.018** |
| Temperature | 1 | 71.99 | 71.99 | 0.14 | 0.715 |
| Treatment | 4 | 1570.90 | 392.73 | 0.75 | 0.562 |
| Population × Temperature | 1 | 4.01 | 4.01 | 0.01 | 0.929 |
| Population × Treatment | 4 | 2797.80 | 699.45 | 1.33 | 0.271 |
| Temperature × Treatment | 4 | 1309.20 | 327.31 | 0.62 | 0.647 |
| Population × Temperature × Treatment | 4 | 515.25 | 128.81 | 0.25 | 0.911 |
| Residual | 60 | 31524 | 525.41 | | |

Significant interactions or main effects are highlighted in bold. df, degrees of freedom; SS, sum of squares; MS, mean sum of squares; Pseudo-F, F value by permutation.

pre-treatment (Fig 5A). At this recovery temperature, the lowest sporophyte recruitment was observed in gametophytes pre-exposed to 22.5˚C. On the other hand, gametophytes pre-exposed to the control, 22.5˚C and 25˚C DHS showed higher sporophyte densities than the 20˚C and darkness treatments during recovery at 15˚C (Fig 5B). These results suggest that pre-conditioning of gametophytes over short time periods and transgression of critical high temperatures negatively influences the subsequent recruitment capacity of *L. digitata* populations. All the interactions showed heterogeneity of dispersions (Table 3), indicating that the significant interactions terms detected in the PERMANOVA could also be due to differences in the dispersion of the data.

The relative presence of female multicellular gametophytes with sporophytes after 20 days of recovery (S2 Fig) showed an overall similar recruitment pattern as the normalized sporophyte density.

### Female and male gametophyte survival

Female gametophyte density after 27 days of recovery differed significantly only due to treatments (Fig 6A; Table 4). It was 1.1-fold reduced by the 25˚C DHS treatment and further 1.3-fold reduced by 22.5˚C compared to the 15˚C control, 20˚C and dark treatments. On the other hand, male gametophyte density after recovery differed only due to population (Table 4; Fig 6B); the Arctic population had higher densities with an average of 224 gametophytes cm$^{-2}$ compared to 212 gametophytes cm$^{-2}$ in the North Sea population (Fig 6B). At the beginning of the experiment, only the combined density of female and male gametophytes was determined since it was difficult to clearly differentiate female and male gametophytes. Therefore, is not possible to know if the population differences in male gametophyte density were already present from the beginning of the experiment.

## Discussion

This study, conducted on individuals from distinct populations but that had lived their whole life in the same conditions, highlights the differential resilience, recovery and reproductive output of microscopic life stages of two populations of *L. digitata* from locations with different long-term thermal histories; Spitsbergen in the Arctic which is near to the northern distribution limit and Helgoland in the North Sea where summer seawater temperatures are comparable or higher than those at the southern distribution limit in Brittany [28, 38]. The responses of gametophytes to temperature treatments suggest a subtle population level adaptive divergence which may be a consequence of their latitudinal distribution gradient. We also demonstrated that the thermal limits were affected by the experimental methodology (static temperatures vs dynamic heat stress) and by the recovery conditions. This mechanistic approach improves our understanding of the response plasticity of a marine kelp species towards temperature stress.

The temperature range for kelp gametophyte growth and reproduction is species-specific and the width ranges between 5 and 19˚C within the genera *Laminaria* and *Saccharina* [25]. In general, seawater temperatures above the broad temperature optimum impair the vegetative growth of gametophytes and their fertilisation success (e.g., [29, 46]). The upper lethal temperature limits after long-term exposure to static temperatures (2 wks exposure) are well known for several selected strains of Atlantic *Laminaria* species [27, 47, 48]. But the effects of dynamic stressful temperature treatments exceeding long-term lethal limits that better mimic natural situations near southern distributional limits has seldom been evaluated in kelps for critical life cycle processes such as growth, reproduction and recruitment of microscopic stages (but see [32]). In our study, gametophytes from both populations exhibited reduced growth rates

compared to the 15˚C control if exposed to continuous sub-lethal temperatures of 22.5˚C or to a dynamic heat stress treatment (25˚C DHS: 3 days of warming from 15˚C to 25˚C, 2 days at peak temperature of 25˚C and 3 days of cooling to 15˚C) irrespective of their original location in the Arctic or North Sea. This uniform response pattern is especially noteworthy as the sub-lethal and dynamic heat stress treatment for the Arctic populations exceeds the local mean summer temperatures by ~15˚C while that of the North Sea population is only exceeded by ~5˚C. This is in line with an earlier study showing that vegetative growth of *L. digitata* gameto-phytes decreased at temperatures of approx. 20˚C [26]. Here, subtle population-specific responses were detected. The North Sea gametophytes from Helgoland showed higher relative growth rates under 15˚C-20˚C and under 25˚C DHS conditions than the Arctic population from Spitsbergen. Given that these cultures are meiotic alternate generation offspring pro-duced and maintained in common culture conditions for an extended period, the maintenance of fixed population differences suggests genetically-based thermal responses linked to their regional origin.

Gametophytes are the most thermally tolerant life cycle stage in kelps, withstanding higher seawater temperatures than sporophytes [27, 30, 47, 49, 50], but upper survival temperature depends on exposure time. The gametophytes of *L. digitata* from Helgoland (North Sea) are able to survive 28˚C for 1 day, 26˚C for 2 days, but this limit already decreases to 24˚C after 7 days and to 23˚C after 4 weeks, thereafter being stable [30]. Similarly, the southern Atlantic species *L. pallida* also shows a 4–5˚C difference in survival temperature between 1 day and 4 weeks exposure [as *L. schinzii*; 30]. In our study, 8 days exposure to 25˚C induced 100% game-tophyte mortality in both populations, highlighting two important aspects: 1) the upper sur-vival temperature in *L. digitata* gametophytes is similar across genetically unconnected populations from contrasting thermal environments and 2) the survival limit within a popula-tion is rather stable within a time period of approx. 40 years, as evidenced by comparing the lethal temperature for Helgoland *L. digitata* gametophyte material sampled in the 1970s (25˚C after 7 days [30]) with the recent material sampled for the current study (25˚C after 8 days). This is despite continuously increasing temperatures within the last decades at this site [37, 51]. However, exposure to lower temperatures (20˚C, 22.5˚C), or to a dynamic heat stress up to lethal 25˚C did not affect the gametophyte density of either population when measured directly after the thermal stress treatments.

Two differential responses only became obvious during a recovery phase at cooler tempera-tures: 1) Female gametophytes were more susceptible to thermal stress than males. While female gametophyte density decreased during recovery from the 25˚C DHS and 22.5˚C treat-ments, male gametophyte density was stable. Higher temperature tolerance of male over female gametophytes thereby seems to be a universal phenotype in kelp species (*L. digitata* [35]; *L. pallida* [= *L. schinzii*] [30]; *Ecklonia* [= *Eckloniopsis*] *radicosa* [52]). 2) Subtle differ-ences in the recovery capacity from thermal stress were observed between populations during gametogenesis and sporophyte development. Overall, the Arctic population displayed slower gametogenesis and lower sporophyte recruitment during recovery from heat stress than the North Sea population. These differences suggest that *L. digitata* gametophytes have adjusted their thermal characteristics in response to regional conditions over evolutionary time scales. Meiospores of *L. digitata* from the Arctic showed high photosynthetic capacity between 7–13˚C, while at 19˚C a significant decrease was observed after only 4h, suggesting lower ther-mal resilience of this Arctic population to high temperatures than temperate populations [53]. In addition, a recent study comparing trailing edge and range centre populations of *L. digitata* sporophytes showed regional differences in the heat shock response suggesting locally adapted thermal ecotypes along a much smaller latitudinal scale [54]. *L. digitata* trailing edge popula-tions were better equipped to tolerate thermal stress. The temperature at which heat shock

proteins were maximally expressed and turned off was 4–8˚C higher compared to range centre populations [54]. Australian *Ecklonia radiata* populations show 1˚C higher optimum temperatures for gametophyte growth from warmer compared to cooler locations [55]. Gametophytes of *Ecklonia radiata* from two New Zealand regions differed in thermal range for growth (warmer location: 9.3˚C—25˚C; cooler location: 8˚C—24˚C) and reproduction (warmer location: 9.3˚C —24˚C; cooler location: < 21.5˚C) [56], while *Ecklonia cava* sporophytes from Japan showed subtle differences in thermal optima for photosynthesis (higher in warmer- than cool-water populations [57]). In contrast, other studies have shown no difference in temperature responses between kelp populations even from different thermal conditions (*L. digitata* [27]; *S. latissima*, *L. digitata*, *S. longicruris* [35]). These results suggest that intraspecific divergence in thermal traits varies among kelp species, but also highlight the importance of studying recovery post-perturbation, as ecologically relevant impacts might only become apparent following a stress event.

In our study recruitment success following thermal treatments was dependent on recovery temperature (5˚C or 15˚C). As in previous studies [29] it was shown that recruitment of *L. digitata* gametophytes was more successful at 5˚C than at 15˚C, in gametophytes that did not experience thermal stress >20˚C. In contrast, recovery at 15˚C enhanced sporophyte development from gametophytes previously exposed to 22.5˚C and 25˚C DHS compared to recovery at 5˚C. Cold temperature recruitment at 5˚C following a summer heat event is unrealistic in nature, but was performed here to allow a mechanistic exploration of recruitment responses within and between populations.

Kelp gametophytes have been reported to play a crucial role as a "seed bank" analogue able to persist in the substrate or as endophytes in a vegetative and thereby dormant stage (e.g., [58– 60]). Postponing gametogenesis during unfavourable environmental conditions for the next life history stage (i.e. sporophytes) and re-establishing reproduction when conditions improve [60] would enhance the probability of successful sporophyte recruitment and growth. Understanding the capacity of gametophytes to recover from extreme thermal events may be critical for population persistence as oceans experience more frequent warming anomalies [3], and since gametophytes can persist and recolonize disturbed areas even after complete sporophyte mortality has occurred [33, 61]. However, in the marine realm the recovery capacity of microscopic life stages of ecologically important foundation species after perturbation remains largely unknown. Our mechanistic approach provides insights into principal recovery mechanisms that may also act in nature. It is however not able to directly correlate results to local *in situ* conditions of the tested populations as the temperature treatments differed relative to local *in situ* summer conditions of the Arctic and Atlantic population. Here, both thermal stress (20˚C, 22.5˚C and 25˚C DHS) and the absence of light prevented the onset of gametogenesis during an exposure of 8 days, while over the same time frame > 45% of female gametophytes from both populations became fertile at the 15˚C control temperature. Although thermally stressed female gametophytes recovered their fertility at 5 and 15˚C recovery temperatures, the subsequent recruitment of new juvenile sporophytes was reduced compared to 15˚C controls, dependent on both the stress and recovery conditions. This corroborates other observations where recruitment capacity returned when dormant microscopic gametophytes experienced suitable conditions again [61, 62–64]. In *M. pyrifera*, gametophytes may delay reproduction for at least seven months when grown under nutrient limiting conditions, even showing enhanced reproductive ability over un-delayed gametophytes when permissive conditions return [63]. Moreover, in the gametophytes of four kelp species (*M. pyrifera*, *Pterygophora californica*, *Laminaria farlowii*, *Pelagophycus porra*), exposure for at least 30 days under unfavourable nutrient conditions conferred a 40%–76% reduction in the time required for sporophyte production after transfer to suitable nutrient conditions [64]. Responses analogous to seed dormancy provide an advantage for populations living in habitats subjected to extreme environmental perturbations. In general,

many species of kelp gametophytes are very resilient to unfavourable conditions including prolonged periods (> 1 yr) of darkness [30]. This capacity of early stages to survive long periods of darkness is an essential ecological strategy to enable juvenile sporophytes to become established beneath dense canopy until light conditions improve [26, 65].

Climate change implies not only a rise in average temperatures, but also an increase in the frequency of extreme thermal events. The number of extremely hot days has increased in coastal areas worldwide over the last three decades [66] and therefore the recovery potential of kelps to extreme heat stress may be crucial for their future persistence [22]. However, studies comparing the effects of both a static and a dynamic temperature shift in a common garden scenario are scarce in macroalgae. Notably, in this study, *L. digitata* gametophytes from both populations showed less tolerance to continuous exposure at 22.5°C than to a dynamic heat stress with a peak temperature of 25°C, with slower rates of gametogenesis and reduced sporophyte production during recovery. This implies that the recovery response is a mixture of the applied temperature duration and its intensity. This complex response towards thermal stress highlights how dynamic environmental conditions in nature might influence the ecological responses of kelps and thereby still limit our capability to precisely predict impacts of future global warming for kelp forest ecosystems. Therefore, studies that better simulate the variable stochastic environmental conditions and that include populations of widespread species from contrasting regional environments are important to better understand future ecosystem dynamics and to sustain conservation and management.

In conclusion, comparative common garden experiments revealed differences in the recovery capacity from thermal stress in microscopic gametophytes of *L. digitata* among two populations from distinct thermal regimes. The magnitude and direction of the response pattern suggest that both populations (representing distinct genetic groups) may have slightly adapted to their local thermal environments since their separation. Continuous exposure to 22.5°C led to a stronger negative impact than gradual increases to a peak temperature of 25°C (lethal for long-term exposure) in gametophytes from both populations. These results highlight the importance of the methodology chosen (static vs dynamic heat stress) in experiments to determine thermal stress responses.

## Supporting information

**S1 Fig. Absolute number of sporophytes after recovery.** Absolute number of sporophytes from the North Sea and Arctic populations of *Laminaria digitata* after 27 days of recovery at 5°C (A) and 15°C (B) from different treatments (15°C dark, 15°C, 20°C, 22.5°C and 25°C DHS). Box plots with median, boxes for 25th and 75th percentiles and whiskers indicating min and max values (n = 4).
(TIF)

**S2 Fig. Percentage of female gametophytes with sporophytes after recovery.** Percentage of female multicellular gametophytes with juvenile sporophytes from the North Sea and Arctic populations of *Laminaria digitata* after 20 days of recovery at 5°C (A) and 15°C (B) from different treatments (15°C dark, 15°C, 20°C, 22.5°C and 25°C DHS). Box plots with median, boxes for 25th and 75th percentiles and whiskers indicating min and max values (n = 4).
(TIF)

## Acknowledgments

We thank D. Liesner and A. Wagner for maintaining the algal material and for help with laboratory facilities.

## Author Contributions

**Conceptualization:** Neusa Martins, Gareth A. Pearson, Ester A. Serrão, Inka Bartsch.

**Data curation:** Neusa Martins.

**Formal analysis:** Neusa Martins.

**Funding acquisition:** Ester A. Serrão.

**Investigation:** Neusa Martins, Julien Bernard.

**Methodology:** Neusa Martins.

**Resources:** Inka Bartsch.

**Supervision:** Gareth A. Pearson, Inka Bartsch.

**Validation:** Gareth A. Pearson, Inka Bartsch.

**Visualization:** Neusa Martins.

**Writing – original draft:** Neusa Martins.

**Writing – review & editing:** Neusa Martins, Gareth A. Pearson, Ester A. Serrão, Inka Bartsch.

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
