## [Decision Letter · Decision Letter 0]

1 Aug 2019

PONE-D-19-17288

Thermal traits for reproduction and recruitment differ between Arctic and Atlantic kelp Laminaria digitata

PLOS ONE

Dear Dr. Martins,

Thank you for submitting your manuscript to PLOS ONE. After careful consideration, we feel that it has merit but does not fully meet PLOS ONE’s publication criteria as it currently stands. Therefore, we invite you to submit a revised version of the manuscript that addresses the points raised during the review process.

Dear Neusa Martins,

With two reviews in hand I am now prepared to recommend this manuscript for a major revision. Both reviewers have raised a number of concerns connected to the experimental design concerning: temperature treatment, statistical analysis i.e. testing data dispersion, and results interpretation.  I would encourage you to respond to the reviewers’ comments point-by-point.  This will make my job easier, and therefore streamline the editorial process.

We would appreciate receiving your revised manuscript by Sep 15 2019 11:59PM. To enhance the reproducibility of your results, we recommend that if applicable you deposit your laboratory protocols in protocols.io, where a protocol can be assigned its own identifier (DOI) such that it can be cited independently in the future. For instructions see: http://journals.plos.org/plosone/s/submission-guidelines#loc-laboratory-protocols

We look forward to receiving your revised manuscript.

Kind regards,

Adrian Zwolicki, Ph.D

Academic Editor

PLOS ONE

Journal Requirements:

2. In your Methods section, please provide additional location information of the collection sites, including geographic coordinates for the data set if available.

Additional Editor Comments (if provided):

Dear Neusa Martins,

With two reviews in hand I am now prepared to recommend this manuscript for a major revision. Both reviewers have raised a number of concerns connected to the experimental design concerning: temperature treatment, statistical analysis i.e. testing data dispersion, and results interpretation. I would encourage you to respond to the reviewers’ comments point-by-point. This will make my job easier, and therefore streamline the editorial process.

Reviewers' comments:

Reviewer's Responses to Questions

**Comments to the Author**

1. Is the manuscript technically sound, and do the data support the conclusions?

Reviewer #1: Partly

Reviewer #2: No

2. Has the statistical analysis been performed appropriately and rigorously? 

Reviewer #1: No

Reviewer #2: No

3. Have the authors made all data underlying the findings in their manuscript fully available?

Reviewer #1: Yes

Reviewer #2: Yes

4. Is the manuscript presented in an intelligible fashion and written in standard English?

Reviewer #1: Yes

Reviewer #2: Yes

5. Review Comments to the Author

Reviewer #1: The manuscript investigates the effect of different warming scenarios on gametophyte density, survival, growth, ontogeny and recruitment of sporophytes from two populations of Laminaria digitata exposed to different thermal regimes. Most research on the impacts of climate change on kelps have been on the macroscopic sporophyte stage so this study adds important information on the microscopic gametophyte stage. I have some issues with the framing of the heatwave experiment as they use the recently developed definition of a marine heatwave proposed by Hobday et al 2016, but then only expose the algae to the highest temperature for 2 days (not a heatwave) and provide no context that this temperature represents a temperature above the 90th percental against a 30 year running mean. Primarily for the former reason (as the authors could provide justification for the later reason) I don’t think the authors should frame this work as testing for the impacts of a marine heatwave. I also think the design lacks ecological realism in places and therefore some of the explanations given need a little more justification. Finally the authors state that where their data failed the assumptions of ANOVA that they used PERMANOVA based on Euclidean distance because this analysis does not need the data to meet assumptions of normality nor homoscedasticity. This is not entirely the case. PERMANOVA is less impacted by data not meeting these assumptions, but the date should generally meet these assumptions to be robust. I suggest the authors run their data through PERMDISP to check these assumptions and if the data is very skewed try to transform the data. If this does not help then many authors take a more conservative approach and reduce the acceptance of significance down to P<0.01.

Minor comments

Line 25 It would be incorrect to state that Helgoland is at the southern distributional range of this species. A more nuanced explanation is provided in the introduction. While this nuanced approach may be too wordy for the abstract I suggest that the authors change this text to perhaps describing that populations were exposed to different thermal regimes.

Line 26 It is not clear what is meant by step-wise here and I suggest that authors consider another term – maybe staggered? The authors need to also mention the continuous dark treatment.

Line 63 Suggest the authors add the following reference which particularly investigates the ecological impacts of MHWs, including for kelps Smale et al (2019) Marine heatwaves threaten global biodiversity and the provision of ecosystem services. Nature Climate Change doi: 10.1038/s41558-019-0412-1

Line 139 While the material used in these experiments were sourced from populations with different thermal regimes they were then cultivated for 3 years and kept at 15 degrees. I think it would be useful for the authors to explain why the thermal regime they were sourced at could explain the differences observed between populations and not some other parameter when they have been kept for 3 years at the same temperature.

Line 162 The authors state that where the temperature was ramped up to 25 degrees over a number of days represented a MHW. I find little ecological rationale to underpin this. The control is 15 degrees and therefore the MHW is 10 degrees above normal, this is an extreme HW when some of the largest observed MHWs have max intensities much lower than this (see Smale et al 2019). At what point in the ramping up of temperature does it reach above the 90th percentile for the region the authors are mimicking? This has impacts on whether this treatment actually meets the definition of a MHW based on a paper that they cite (Hobday et al 2016), which explicitly states a MHW must be above the 90th percentile for a period of at least 5 days.

Line 164 The treatment was unlikely a MHW for the full 8 days – see comment above.

Line 167 The dark treatment is suggested to mimic conditions under a dense canopy. I have dived in locations with very dense canopies on very overcast days and it is never full darkness. Can the authors provide evidence that this treatment does represent the conditions that they state.

Lines 174-176 It is stated the 16:8 light:dark settings are chosen to reflect summer conditions, but for the northern population light hours would be longer than this. I suggest the authors clarify this statement.

Line 188-189 The authors state that 5 and 15 degree temperatures were chosen as recovery temperatures because they reflected mean winter and summer temperatures respectfully. I find little ecological realism in the idea that SST could be a 25 degrees and then drop to 5 degrees over the time scales of this study. This treatment lacks ecological realism and I think the authors need to justify its inclusion.

Table 1, 2, 3 & 4- In the legend state the test performed

Gametophyte growth – the authors needs to state how the dark treatment affected growth

Lines 304-311 At present the text is based on a qualitative look at the data. I believe it would be better if this data was analysed quantitatively to really show where the differences lay.

Line 310 I got a little confused with what time point is being referred at 8 days. Is this in fact day 0 or recovery or day 8 of recovery and if the later how is the reader able to agree with the authors when figure 4 is plotted as day 0, 5, 10 etc. I suggest that this text needs some clarification.

Line 321 This starts with in general, when this pattern looks across the board on my interpretation of the figure. Suggest rephrasing.

Line 331-333 This statement isn’t true for the North Sea population where sporophyte densities were the same or higher in the 15 degree recovery treatment at 22.5 and 25 degrees. This is stated in the next sentence, but I suggest this section is restructured to avoid confusion.

Table 3 – suggest choosing a more conservative p-value if the data fails the PERMDISP test

Female and male gametophyte survival – no mention is made of the dark treatment

Line 398 I suggest the readers use more nuanced language as they did in the introduction regarding Helgoland being the southern distributional boundary of L. digitata

Line 399 – 401 I am not convinced by this statement. For density, with the exception of continuous 25 degrees there was no effect of treatment or source population after 8 days. After 27 days there was a treatment effect on female gametophyte density, but this was not related to the source population, while for male gametophyte density there was a population effect, but this did not interact with temperature. No stats were run on the ontogeny data. Therefore the only response variables that had a treatment x population interaction which would be required to state that there was population level adaptive divergence along a latitudinal (i.e. temperature) gradient are for growth and sporophyte recruitment. I therefore believe this statement needs to be a little more nuanced.

Line 404 I am not sure how this study improves our understanding of the genetic potential for recovery. I suggest the authors provide a more detailed explanation here.

Lines 410-411 I am not clear on what is trying to be said here and suggests the authors rephrase for clarity

Line 416 Again a little more clarity is required here. State which 25 degree treatment you are referring to here as both did not lead to mortality

Line 448-449 This was not the case for all response variables and the more nuanced response needs to be described.

Line 516 – 519 I suggest as well as ramp up of temperature that exposure duration could also have led to the differential results and should be mentioned. These differences not only are likely to influence species thermal limits, but the way we run experiments and whether we include acclimation at different temperatures or just a heatshock will also influence our interpretation of the likely future impacts of warming and perhaps this also deserves a mention.

Reviewer #2: Martins and co-worker investigated the thermal traits (and tolerance) for gametogenesis, sexual reproduction and sporophyte recruitment of Arctic and North Sea Laminaria digitata. The experiments were conducted in the laboratory in a common garden set-up. However, the experimental temperature treatment was clearly biased towards the southern population; disregarding the fact that temperature adaptation and history of the two populations are different. In this regard, the authors concluded that gametophytes from the North Sea exhibited higher growth rates and greater sporophyte recruitment after thermal stress compared with the Arctic; which can be indirectly interpreted as that North Sea population is better adapted to thermal stress compared to Arctic population. This is contrary to the data presented and the conclusion was based on missed logical interpretation. Moreover, the suggestion that thermal characteristics of the two populations diverge over evolutionary time scales is speculative and not supported by the data presented.

Considering that the summer high temperature is 5-6°C in the Arctic and 18°C (or higher) in the North Sea, the temperature treatment of 15, 20, 22.5, and 25°C in the common garden experiment are effectively in the range of 10-20°C and 2-7°C increase in temperature for Arctic and North Sea populations, respectively. Relative to the higher magnitude of temperature increase compared to the respective in situ summer high temperature experienced by the corresponding populations, data suggest that Arctic population has higher tolerance to thermal stress compared to the North Sea population.

For example, in Figure 3. Without treatment and data at temperature lower than 15°C, data suggest that Arctic population is more tolerant to thermal stress because growth rate between 15 and 20°C is not significantly different.

Had there been temperature treatment lower than 15°C for the Arctic population, two scenarios are possible:

1. If at temp < 15°C (e.g. 5 and 10°C), growth rate could be equal to 15 and 20°C. Therefore, Arctic population have higher tolerance to thermal stress. This is equal to max. 15°C change in temperature.

2. If at temp < 15°C (e.g. 5 and 10°C), growth rate is ×-fold higher than at 15 and 20°C, then Arctic population is more sensitive to thermal stress.

A third scenario is possible:

3. If at temp < 15°C (e.g. 5 and 10°C), growth rate is ×-fold lower than at 15 and 20°C. How will this change the conclusion?

On the other hand, growth rate of Helgoland population (which experience 18°C summer high temperature) already had significant decline from 15 to 20°C, which is only 5°C change in temperature. Therefore, the population is more sensitive to temperature change (population is living on the edge!). Had there been an 18°C treatment, would growth rate had been higher or lower compared to 15°C? How will this change the conclusion?

The above are hypothetical but pertinent questions, which should have been considered in the design of the experiments.

Pairwise comparison between populations under the same temperature (e.g. Arctic vs. North Sea at 15 and 20°C) is meaningless because the summer high temperatures experienced by the two populations between populations are different such that at 15°C, Artic population encountered 10°C increase in temperature while the North Sea population encountered 3°C decrease in temperature.

The same is with data and statistical analysis in Fig. 5. Pairwise comparison doesn't make sense.

For North Sea population, the temperature increase from 18°C summer high temperature to 22.5- 25°C is max. 7°C; while for Arctic population, the temperature increase from 5-6°C summer high temperature to 22.5 -25°C is max. 20°C. Naturally, the Arctic population will have lower recovery rate compared to Helgoland regardless of recovery temperature. The authors need to reassess their experimental design and data interpretation and consider a paradigm shift. Data and statistical analyses, results and discussion will substantially change accordingly.

What is the relevance of dark control? Without light (or under very low light) growth will naturally be arrested.

Minor comments:

Lines 85-86: How about the collapse of Saccharina population in south and west coast of Norway Norway (Moy and Christie 2012)?

Line 107-110: Example of large-scale disturbance? For example, storm causing large scale dislodgment of adult kelp sporophytes (Roleda and Dethleff 2011) in Helgoland. However, recovery and establishment of new generation of recruits are dependent on the seed bank (e.g. Hoffmann and Santelices 1991).

Line 129: Please provide respective collection dates.

Lines 145-156: Please clarify preparation of stock solution and replication. What was mixed? Males and females from 5 different individual were separately mixed to obtain stock solutions of (1) male and (1) female? How was the replication (n=4) done?

Line 159: Why control? What is control here?

Line 157-170: Justify was start treatment at 15°C and disregarded the in-situ summer high temperature in the Arctic. Ideally, should have additional lower temperature treatment at 5 and 10°C for both populations. Why employ 5°C temperature for recovery only?

Line 185-190: Arctic population treatment start at 15°C and allowed to recovery at 5°C. Why use the summer high temperature for recovery? On the other hand, why let Helgoland population recover at 5°C, which is nowhere near the summer high temperature? Subsequently described as the winter low temperature for the southern population. There seems to be great disparity in the handling of experimental treatment and recovery. Which southern limit of Ldig population (where) experiences a winter temp of 5°C? Is there any ecological relevance in the treatment of the two different populations?

Lines 234-239: What is female cell per vegetative gametophyte? Are not all cells in the female gametophyte females? Wasn’t gametophytes’ length and density were standardized at the beginning. Then equal volume of stock male and female gametophytes were supposedly mixed in every population (Lines 146-154). Was this not temperature effect? Or artifact? How did normalization solved the problem?

Line 257: The relevance of posthoc pairwise t-test is in question.

Line 276-281 (Figure 2): Why would gametophyte density change when "seeding" at the beginning was already controlled (Lines 146-154)?

Figure 5 and Figure 6: Which comes first? The survival of male and female gametophytes (Fig. 6) or the fertilization and production of embryonic and juvenile sporophytes (Fig. 5)? Data were obtained from the same experimental units?

Please consider the following literatures:

Roleda 2009. Photosynthetic response of Arctic kelp zoospores exposed to radiation and thermal stress. Study showed that photosynthetic efficiency of Arctic Ldig under 2, 7 and 13°C did not change within 48h period, but slowly declined at 19°C.

Liu et al. 2017. Seaweed reproductive biology: environmental and genetic controls

6. PLOS authors have the option to publish the peer review history of their article (what does this mean?). If published, this will include your full peer review and any attached files.

Reviewer #1: No

Reviewer #2: No

---

## [Author Response · Author response to Decision Letter 0]

12 Nov 2019

Answers to Reviewers' comments:

Reviewer #1

The manuscript investigates the effect of different warming scenarios on gametophyte density, survival, growth, ontogeny and recruitment of sporophytes from two populations of Laminaria digitata exposed to different thermal regimes. Most research on the impacts of climate change on kelps have been on the macroscopic sporophyte stage so this study adds important information on the microscopic gametophyte stage. I have some issues with the framing of the heatwave experiment as they use the recently developed definition of a marine heatwave proposed by Hobday et al 2016, but then only expose the algae to the highest temperature for 2 days (not a heatwave) and provide no context that this temperature represents a temperature above the 90th percental against a 30 year running mean. Primarily for the former reason (as the authors could provide justification for the later reason) I don’t think the authors should frame this work as testing for the impacts of a marine heatwave. 

We appreciated the Reviewer’s suggestions and comments. We agree that this does not correspond to the heatwave definition of Hobday et al. (2016) therefore we now call it dynamic heat stress rather than heat wave. The important point is to make clear what we are testing. We improved our explanation of the research question in the text to clarify that the objective is to investigate population differences in functional traits, in this case by asking whether there are population differences in thermal tolerance and performance. As populations are less likely to differ under optimal conditions and indeed pre-investigations showed that thermal optima within different geographical isolates of this species seem to be quite stable (Bolton and Luning 1982, tom Dieck 1992), we used sub-lethal and lethal temperature conditions. The goal of the paper has now been clarified (Line 157).

I also think the design lacks ecological realism in places and therefore some of the explanations given need a little more justification. 

We used a mechanistic approach to investigate population differences in functional traits under a set of extreme thermal conditions and not to mimic in situ current and future environmental conditions. Thus, in this new version of the manuscript, we decided to reduce the ecological relevance sentences that were diverting attention from the main aim of the study. 

Finally the authors state that where their data failed the assumptions of ANOVA that they used PERMANOVA based on Euclidean distance because this analysis does not need the data to meet assumptions of normality nor homoscedasticity. This is not entirely the case. PERMANOVA is less impacted by data not meeting these assumptions, but the date should generally meet these assumptions to be robust. I suggest the authors run their data through PERMDISP to check these assumptions and if the data is very skewed try to transform the data. If this does not help then many authors take a more conservative approach and reduce the acceptance of significance down to P<0.01.

We appreciated the reviewer comment. As suggested, our data were checked for homogeneity of multivariate dispersions through PERMDISP. Data were then transformed using log, square-root or fourth-root transformations, however the heterogeneity of variances and the skewed distribution persisted, and therefore non-transformed data were used. Because the dispersion tests were significant, a more conservative p-value (P<0.01) was used. We inserted this information in the Materials and methods section (Line 317) and changed the Results (Line 402) accordingly. 

Minor comments

• Line 25 It would be incorrect to state that Helgoland is at the southern distributional range of this species. A more nuanced explanation is provided in the introduction. While this nuanced approach may be too wordy for the abstract I suggest that the authors change this text to perhaps describing that populations were exposed to different thermal regimes.

We changed the text as suggested, now describing that the populations used in this study originate from distinct thermal regimes (Line 25).

• Line 26 It is not clear what is meant by step-wise here and I suggest that authors consider another term – maybe staggered? The authors need to also mention the continuous dark treatment.

As suggested, we changed the term stepwise heatwave into dynamic thermal treatment and mentioned the continuous dark treatment (Line 28).

• Line 63 Suggest the authors add the following reference which particularly investigates the ecological impacts of MHWs, including for kelps Smale et al (2019) Marine heatwaves threaten global biodiversity and the provision of ecosystem services. Nature Climate Change doi: 10.1038/s41558-019-0412-1

Following the previous reviewer’s comments regarding the heat wave definition, we decided to rename our temperature-ramped treatment (previously heatwave) to dynamic heat stress treatment and to reduce the introductory sentences concerning the definition, description and ecological consequences of MHWs in order to improve the focus and the clarity of the manuscript. In this new version of the manuscript, in the introduction we focused more on population differences in thermal response, which is the main aim of this study. For this reason, the reference Smale et al. (2019) dealing with the ecological impacts of MHWs was not included. 

• Line 139 While the material used in these experiments were sourced from populations with different thermal regimes, they were then cultivated for 3 years and kept at 15 degrees. I think it would be useful for the authors to explain why the thermal regime they were sourced at could explain the differences observed between populations and not some other parameter when they have been kept for 3 years at the same temperature.

The cultures used in the experiments are microscopic haploid offspring (result from meiosis) from field-collected kelp, produced under the same culture conditions (irradiance, temperature, daylength, salinity, etc) and having never seen distinct conditions during their life time. Therefore, other than possible maternal effects, the population differences observed can be attributed to genetically-based differences in thermal responses linked to their population of origin, because no other factors differed during culture conditions for 3 years. This information was included in the manuscript (Line 160). 

• Line 162 The authors state that where the temperature was ramped up to 25 degrees over a number of days represented a MHW. I find little ecological rationale to underpin this. The control is 15 degrees and therefore the MHW is 10 degrees above normal, this is an extreme HW when some of the largest observed MHWs have max intensities much lower than this (see Smale et al 2019). At what point in the ramping up of temperature does it reach above the 90th percentile for the region the authors are mimicking? This has impacts on whether this treatment actually meets the definition of a MHW based on a paper that they cite (Hobday et al 2016), which explicitly states a MHW must be above the 90th percentile for a period of at least 5 days.

Please see reply above. This comment is talking about a research question that is not the question of this paper. We have therefore clarified the goals of the paper in the introduction and removed the heat wave rationale as defined by Hobday et al. (2016) because this was not at all the objective of the paper. We did not clearly point this out in the first version. We agree that this does not correspond to the heatwave definition of Hobday et al. (2016), therefore we now call it dynamic heat stress rather than heat wave. The important point is to make clear what we are testing, which is whether there are conditions under which populations show differences in fitness responses to any particular temperature conditions. We improved our explanation of the research question in the text to clarify that the objective is to investigate population differences in functional traits in a mechanistic approach, in this case by asking whether there are population differences in thermal tolerance and performance after heat stress. 

• Line 164 The treatment was unlikely a MHW for the full 8 days – see comment above.

We agree that the ramped thermal treatment used in this study does not follow the MHW definition of Hobday et al. (2016). Since the objective of the paper was not to apply a heat wave with that definition, we rename this thermal treatment a dynamic heat stress. Please see also reply above.

• Line 167 The dark treatment is suggested to mimic conditions under a dense canopy. I have dived in locations with very dense canopies on very overcast days and it is never full darkness. Can the authors provide evidence that this treatment does represent the conditions that they state.

We decided to use the dark treatment to mimic the very low light conditions that the gametophytes are exposed in some particular conditions (below overhangs or under stipes, covered by sediment, etc). In situ measurements in the Arctic under a very dense kelp canopy show that irradiance at the substrate surface is near zero, even in sunny and clear-sky summer days that are extremely rare at Kongsfjorden (Please see Fig. 2 in Laeseke et al. 2019; Pavlov et al. 2019). We add this information in the manuscript (Line 223). In addition, algae occurring at lower latitudes and in the winter in the Arctic where they can be exposed to extended periods of darkness due to polar nights and sea ice covering, the irradiance can be even lower or zero. In fact, if the ice is also covered by snow, light can be decreased to < 2% of the light values in the surface, and therefore they may be exposed to 10 months of darkness or very low light conditions (please see review Wiencke et al. 2006). Nevertheless, we agree that darkness is physiologically different to extremely low light conditions. However, this study also aim to compare the response of gametophytes from the Arctic and the North Sea populations under darkness conditions to check if the Arctic gametophytes recover better from the absence of light than the North Sea gametophytes.

• Lines 174-176 It is stated the 16:8 light:dark settings are chosen to reflect summer conditions, but for the northern population light hours would be longer than this. I suggest the authors clarify this statement.

We chose the 16:8h light:dark regime for both populations as it is a good approximation for summer daylengths along the whole distributional range of the species. It has long been known (see Breeman and Guiry 1989) that seasons in the subtidal are different from above the sea level. Recently it was shown for a kelp bed in the Arctic (Pavlov et al. 2019, Fig. 5.6 and Laeseke et al. 2019) that even during summer, below dense kelp canopies, the effective light level is below 24:0h light:dark. In addition, a continuous light regime may induce a completely altered physiology and thereby may induce even more uncontrolled processes. Lüning (1981) has shown that while egg release in kelp gametophytes is diurnal, they are released from this diurnal pattern under continuous irradiance. Moreover, there is a wealth of literature showing that many physiological responses act in a diurnal way (e.g., photosynthesis). This may have influenced our thermal experiment by introducing factors over which we have little control, thus reducing our ability to detect thermal effects. We clarified the reason for using the 16:8h light:dark regime in the Materials and methods section (Line 233).

• Line 188-189 The authors state that 5 and 15 degree temperatures were chosen as recovery temperatures because they reflected mean winter and summer temperatures respectfully. I find little ecological realism in the idea that SST could be a 25 degrees and then drop to 5 degrees over the time scales of this study. This treatment lacks ecological realism and I think the authors need to justify its inclusion.

Please see reply above. Yes, the experimental design is a mechanistic approach to investigate the potential adaptive capacity of the two populations under thermal regimes that could theoretically be experienced by the meta-species. We were interested how the two populations might fit into this scheme. We made it clearer now that we did not simulate immediate ecological conditions. A delta treatment investigation would not really provide information whether the adaptive potential of populations which have evolved with contrasting thermal histories, is different. Pre-information on L. digitata has shown that population differences with respect to thermal responses are subtle, thus we used temperature conditions in which they might differ. We removed the ecological argument for choosing the recovery temperatures of 5ºC and 15ºC.

• Table 1, 2, 3 & 4- In the legend state the test performed

The suggestion of the reviewer was accepted and we have included the test performed in the legend of the statistical Tables.

• Gametophyte growth – the authors needs to state how the dark treatment affected growth

As suggested, we inserted how the dark treatment during 8 days affected gametophyte growth in the results section (Line 355).

• Lines 304-311 At present the text is based on a qualitative look at the data. I believe it would be better if this data was analysed quantitatively to really show where the differences lay.

The text concerning the development of the different ontogenetic stages in the results section is in fact based on quantitative data and not merely qualitative data. We quantified in female gametophytes the percentage of four ontogenetic stages: vegetative stage, gametophytes with oogonia, gametophytes with eggs released and gametophytes with sporophytes attached (Please see line 278 in the Materials and methods section) and the results description are based on these quantitative data.

• Line 310 I got a little confused with what time point is being referred at 8 days. Is this in fact day 0 or recovery or day 8 of recovery and if the later how is the reader able to agree with the authors when figure 4 is plotted as day 0, 5, 10 etc. I suggest that this text needs some clarification.

In the text the time point referred as day 8 was at the end of the period of darkness and heat stress so it was the day 0 of the recovery phase; this was clarified in the manuscript (Line 380). In Figure 4, only the recovery periods are shown (day 0, 5, 10, 15, 20) and the x axis is clearly labelled as “Recovery time (days)”, so no changes were made in the figure, however we clarified this information in the figure legend.

• Line 321 This starts with in general, when this pattern looks across the board on my interpretation of the figure. Suggest rephrasing.

In both populations, lower proportions of female gametophytes were fertile at the recovery temperature of 5ºC compared to 15ºC after almost all treatments (15ºC dark, 15ºC, 20ºC, 22.5ºC and 25ºC DHS) with the exception of the 15°C control temperature. The female gametophytes exposed to the 15°C control exhibited the fastest fertility at the recovery temperature of 5ºC. We changed the sentence for clarity (Line 393).

• Line 331-333 This statement isn’t true for the North Sea population where sporophyte densities were the same or higher in the 15 degree recovery treatment at 22.5 and 25 degrees. This is stated in the next sentence, but I suggest this section is restructured to avoid confusion.

The reviewer suggestion was accepted, we restructured the text for clarity (Line 403). 

• Table 3 – suggest choosing a more conservative p-value if the data fails the PERMDISP test

In accordance with the reviewer comment, a more conservative p-value of 0.01 was adopted as PERMDISP tests were significant (Line 436).

• Female and male gametophyte survival – no mention is made of the dark treatment

The suggestion of the reviewer was accepted and we have inserted the effect of the dark treatment on the female gametophyte survival (Line 466). The male gametophyte survival was not affected by the different treatments and this information is already mentioned in the text (Line 467).

• Line 398 I suggest the readers use more nuanced language as they did in the introduction regarding Helgoland being the southern distributional boundary of L. digitata

In accordance with the reviewer comment, we changed the sentence to make it clear that Helgoland is not the southern distributional boundary but the region where summer seawater temperatures may be comparable or higher to those at the southern distribution limit (Line 494). 

• Line 399 – 401 I am not convinced by this statement. For density, with the exception of continuous 25 degrees there was no effect of treatment or source population after 8 days. After 27 days there was a treatment effect on female gametophyte density, but this was not related to the source population, while for male gametophyte density there was a population effect, but this did not interact with temperature. No stats were run on the ontogeny data. Therefore the only response variables that had a treatment x population interaction which would be required to state that there was population level adaptive divergence along a latitudinal (i.e. temperature) gradient are for growth and sporophyte recruitment. I therefore believe this statement needs to be a little more nuanced.

As suggested we modified the statement so that the subtle adaptive divergence found among the two populations of L. digitata from contrasting thermal environments is not so much emphasized (Line 497).

• Line 404 I am not sure how this study improves our understanding of the genetic potential for recovery. I suggest the authors provide a more detailed explanation here.

The reviewer comment was accepted. The statement was removed from the manuscript.

• Lines 410-411 I am not clear on what is trying to be said here and suggests the authors rephrase for clarity

We recognize that this sentence was not clear and, therefore, we clarified in the manuscript (Line 509).

• Line 416 Again a little more clarity is required here. State which 25 degree treatment you are referring to here as both did not lead to mortality

The 25ºC treatment mentioned in the sentence was the dynamic heat stress treatment, where the temperature was increased from 15ºC to 25ºC at a warming rate of 2-3ºC day-1 and the temperature of 25ºC was kept over a period of 2 days. Afterwards, the temperature was decreased from 25ºC back to 15ºC at the same rate. We clarified this information in the manuscript (Line 518). 

• Line 448-449 This was not the case for all response variables and the more nuanced response needs to be described.

Indeed, during the recovery stage not all the response variables showed differences between populations to thermal stress. Differences were detected during the gametogenesis development and in the sporophyte recruitment. However, the response variables: female and male gametophyte density did not show a treatment × population interaction during the recovery stage. Therefore, we agree with the reviewer that during the recovery stage not all the response variables showed differences between populations to the treatments under study and that the observed differences in the fertility and recruitment were subtle, however this was explained in detail in the text (please see line 555). 

• Line 516 – 519 I suggest as well as ramp up of temperature that exposure duration could also have led to the differential results and should be mentioned. These differences not only are likely to influence species thermal limits, but the way we run experiments and whether we include acclimation at different temperatures or just a heatshock will also influence our interpretation of the likely future impacts of warming and perhaps this also deserves a mention.

We thank the reviewer for the valuable suggestions, and we have considered the temperature exposure duration for the different results obtained at the continuous temperature of 22.5ºC and at the dynamic heat stress (25ºC DHS) with a peak temperature of 25ºC (Line 642). We also mentioned that the experimental design not only influences the thermal limits but also the interpretation of the future global warming impacts (Line 647).

Reviewer #2: 

Martins and co-worker investigated the thermal traits (and tolerance) for gametogenesis, sexual reproduction and sporophyte recruitment of Arctic and North Sea Laminaria digitata. The experiments were conducted in the laboratory in a common garden set-up. However, the experimental temperature treatment was clearly biased towards the southern population; disregarding the fact that temperature adaptation and history of the two populations are different. In this regard, the authors concluded that gametophytes from the North Sea exhibited higher growth rates and greater sporophyte recruitment after thermal stress compared with the Arctic; which can be indirectly interpreted as that North Sea population is better adapted to thermal stress compared to Arctic population. This is contrary to the data presented and the conclusion was based on missed logical interpretation. Moreover, the suggestion that thermal characteristics of the two populations diverge over evolutionary time scales is speculative and not supported by the data presented.

Considering that the summer high temperature is 5-6°C in the Arctic and 18°C (or higher) in the North Sea, the temperature treatment of 15, 20, 22.5, and 25°C in the common garden experiment are effectively in the range of 10-20°C and 2-7°C increase in temperature for Arctic and North Sea populations, respectively. Relative to the higher magnitude of temperature increase compared to the respective in situ summer high temperature experienced by the corresponding populations, data suggest that Arctic population has higher tolerance to thermal stress compared to the North Sea population.

The comments of the referee are correct but that is not the research question of this paper. This paper does not aim to investigate whether populations will survive x degrees above their summer temperature. It aims to investigate whether this species has distinct thermal tolerances depending on the population of that same species; i.e., whether distinct populations of the same species have any functional differences at any temperatures. It is not an ecological question, it is an evolutionary question. Therefore, we clarified the aim of this study throughout the manuscript.

For example, in Figure 3. Without treatment and data at temperature lower than 15°C, data suggest that Arctic population is more tolerant to thermal stress because growth rate between 15 and 20°C is not significantly different.

Had there been temperature treatment lower than 15°C for the Arctic population, two scenarios are possible:

1. If at temp < 15°C (e.g. 5 and 10°C), growth rate could be equal to 15 and 20°C. Therefore, Arctic population have higher tolerance to thermal stress. This is equal to max. 15°C change in temperature.

2. If at temp < 15°C (e.g. 5 and 10°C), growth rate is ×-fold higher than at 15 and 20°C, then Arctic population is more sensitive to thermal stress.

A third scenario is possible:

3. If at temp < 15°C (e.g. 5 and 10°C), growth rate is ×-fold lower than at 15 and 20°C. How will this change the conclusion?

On the other hand, growth rate of Helgoland population (which experience 18°C summer high temperature) already had significant decline from 15 to 20°C, which is only 5°C change in temperature. Therefore, the population is more sensitive to temperature change (population is living on the edge!). Had there been an 18°C treatment, would growth rate had been higher or lower compared to 15°C? How will this change the conclusion?

The above are hypothetical but pertinent questions, which should have been considered in the design of the experiments.

We appreciated the reviewer comment; however this comment is talking about a research question that is not the question of this paper. 

Pairwise comparison between populations under the same temperature (e.g. Arctic vs. North Sea at 15 and 20°C) is meaningless because the summer high temperatures experienced by the two populations between populations are different such that at 15°C, Artic population encountered 10°C increase in temperature while the North Sea population encountered 3°C decrease in temperature.

The same is with data and statistical analysis in Fig. 5. Pairwise comparison doesn't make sense.

For North Sea population, the temperature increase from 18°C summer high temperature to 22.5- 25°C is max. 7°C; while for Arctic population, the temperature increase from 5-6°C summer high temperature to 22.5 -25°C is max. 20°C. Naturally, the Arctic population will have lower recovery rate compared to Helgoland regardless of recovery temperature. The authors need to reassess their experimental design and data interpretation and consider a paradigm shift. Data and statistical analyses, results and discussion will substantially change accordingly.

This comment is talking about a research question that is not the question of this paper. In this study, pairwise comparison are appropriate because we aim to investigate population differences in functional traits, by asking whether there are population differences in thermal tolerance under any particular conditions in which they might differ.

What is the relevance of dark control? Without light (or under very low light) growth will naturally be arrested.

The dark treatment was used as a non-optimal environmental condition for gametophyte growth and reproduction. It can simulate microscopic gametophytes growing in the very shaded sub-canopy under parental sporophytes (below overhangs or under stipes, covered by sediment, etc). In fact, in situ measurements in the Arctic under a very dense kelp canopy show that irradiance at the substrate surface is near zero, even in sunny and clear-sky summer days that are extremely rare at Kongsfjorden (Please see Fig. 2 in Laeseke et al. 2019; Pavlov et al. 2019). This information was added in the manuscript (Line 223). Irradiance can be even lower (near zero) for algae occurring at lower latitudes and for algae occurring in the Arctic it can also simulate the extended periods of darkness due to polar nights and sea ice covering that they are exposed during winter. This treatment was used to investigate if the gametophytes are able to become fertile and develop sporophytes after being exposed to optimum light conditions. Therefore, the main aim of using the dark treatment was not to check the effect on the gametophyte growth, but to evaluate the recovery capacity of the gametophytes after a non-inductive gametogenesis environmental condition, but also non-lethal. 

Minor comments:

• Lines 85-86: How about the collapse of Saccharina population in south and west coast of Norway Norway (Moy and Christie 2012)?

The suggestion of the reviewer was accepted and we have inserted the work on the collapse of Saccharina latissima in the coast of Norway (Line 126). 

• Line 107-110: Example of large-scale disturbance? For example, storm causing large scale dislodgment of adult kelp sporophytes (Roleda and Dethleff 2011) in Helgoland. However, recovery and establishment of new generation of recruits are dependent on the seed bank (e.g. Hoffmann and Santelices 1991).

As recommended, the work of Roleda and Dethleff (2011) on the large scale dislodgment of kelp sporophytes in Helgoland was taken into consideration and inserted (Line 146) in the manuscript as well as the work of Hoffmann and Santelices (1991).

• Line 129: Please provide respective collection dates.

As suggested, we have inserted the respective collection dates for each site (Line 180).

• Lines 145-156: Please clarify preparation of stock solution and replication. What was mixed? Males and females from 5 different individual were separately mixed to obtain stock solutions of (1) male and (1) female? How was the replication (n=4) done?

Yes, male and female gametophytes from 5 sporophytes per population were mixed separately to obtain 4 stock solutions (Helgoland ♀; Helgoland ♂; Spitsbergen ♀; Spitsbergen ♂), i.e., each stock solution had a mix of 5 strains of gametophytes. 

The density of each stock solution was calculated, female and male from each population were mixed to have equal proportions of both sexes. The combined solution was distributed to Petri dishes, each with ~400 gametophytes cm-2. Four Petri dishes were used per population and treatment as already described in the text (Line 207), thus we defined as replicates the Petri dishes. 

As recommended, we clarified the preparation of the stock solution and replication in the Materials and Methods section (Line 204, 208).

• Line 159: Why control? What is control here?

We considered the temperature of 15ºC as the control because it is within the optimal temperature range for the growth and reproduction of L. digitata gametophytes (Lüning 1980, tom Dieck 1992, Martins et al. 2017). Moreover, the gametophytes from both populations were healthily growing vegetatively at 15ºC in the laboratory prior to the experiment, so it has been their growth temperature for years. This information was added to the Materials and methods section (Line 224).

• Line 157-170: Justify was start treatment at 15°C and disregarded the in-situ summer high temperature in the Arctic. Ideally, should have additional lower temperature treatment at 5 and 10°C for both populations. Why employ 5°C temperature for recovery only?

As already explained above, the aim of this work is to investigate genetically-based population differences in functional traits, by asking whether there are population differences in thermal tolerance under any particular conditions in which they might differ despite having spent their entire lives in the same conditions.

• Line 185-190: Arctic population treatment start at 15°C and allowed to recovery at 5°C. Why use the summer high temperature for recovery? On the other hand, why let Helgoland population recover at 5°C, which is nowhere near the summer high temperature? Subsequently described as the winter low temperature for the southern population. There seems to be great disparity in the handling of experimental treatment and recovery. Which southern limit of Ldig population (where) experiences a winter temp of 5°C? Is there any ecological relevance in the treatment of the two different populations?

As explained above the aim of the study was to investigate whether populations from thermally contrasting environments of the same species have differences in the reproductive performance of microscopic gametophytes and the sporophyte recruitment capacity in any temperature. We let both populations recover at 5ºC and 15ºC to check for differences in the recover capacity under cold and warm temperatures. To clarify the focus of this study, we removed the ecological argument for the chosen recovery temperatures from the manuscript.

• Lines 234-239: What is female cell per vegetative gametophyte? Are not all cells in the female gametophyte females? Wasn’t gametophytes’ length and density were standardized at the beginning. Then equal volume of stock male and female gametophytes were supposedly mixed in every population (Lines 146-154). Was this not temperature effect? Or artifact? How did normalization solved the problem?

Each female vegetative gametophyte had several cells and the number of cells per gametophyte fragment (called female in the MS) differed slightly due to the crushing procedure. We understand that our sentence was confusing and, therefore, we changed in the manuscript (Line 297). Indeed, in the beginning of the experiment we did our best to control the number of cells per gametophyte fragment in both populations by standardizing the density to ~400 gametophytes cm-2 and the length of the gametophytes to 50-100 µm by sieving them. Anyway, the number of cells per female gametophyte fragment can still be slightly different between 50-100 µm. As the Arctic population produced higher number of sporophytes per cm2 compared to the North Sea population at 15ºC that is an optimum temperature to produce sporophytes in L. digitata (Martins et al. 2017), this may indicate differences in the initial number of cells per female gametophyte fragment. As strong differences were observed regarding the number of sporophytes between populations under optimum conditions, we decided to normalize the data in order to be possible to evaluate the effects of the different treatments. 

• Line 257: The relevance of posthoc pairwise t-test is in question.

Post-hoc pair-wise comparisons were performed whenever a significant difference between treatments/populations was found. As the aim of this study was to compare the recovery capacity to recruit of distinct populations of the same species in any specific temperature, the Pos-hoc pair-wise comparisons used were valid.

• Line 276-281 (Figure 2): Why would gametophyte density change when "seeding" at the beginning was already controlled (Lines 146-154)?

Indeed, gametophyte density was controlled at the beginning of the experiment; however it can decrease due to gametophyte death after exposure to lethal thermal treatments. In this study, the gametophytes density was zero after 8 days of exposure to the continuous temperature of 25ºC, meaning that no gametophytes survived under this thermal treatment.

• Figure 5 and Figure 6: Which comes first? The survival of male and female gametophytes (Fig. 6) or the fertilization and production of embryonic and juvenile sporophytes (Fig. 5)? Data were obtained from the same experimental units?

Although they were independent measures, the absolute number of sporophytes (Fig. 5) and the survival of male and female gametophytes (Fig. 6) were both measured after 27 days of recovery as described in the Figure legends (Line 423 and 478). Since, gametogenesis (development of ontogenetic stages over recovery time represented in Fig. 4) and recruitment (sporophyte density data showed in Fig. 5) are biological processes that occur sequentially, we decided to present their results in the same order. And by last the survival of male and female gametophytes (Fig. 6).

Please consider the following literatures:

Roleda 2009. Photosynthetic response of Arctic kelp zoospores exposed to radiation and thermal stress. Study showed that photosynthetic efficiency of Arctic Ldig under 2, 7 and 13°C did not change within 48h period, but slowly declined at 19°C.

Liu et al. 2017. Seaweed reproductive biology: environmental and genetic controls

As recommended, the suggested citations were included in the manuscript.

---

## [Decision Letter · Decision Letter 1]

10 Jan 2020

PONE-D-19-17288R1

Thermal traits for reproduction and recruitment differ between Arctic and Atlantic kelp Laminaria digitata

PLOS ONE

Dear Dr. Martins,

Thank you for submitting your manuscript to PLOS ONE. After careful consideration, we feel that it has merit but does not fully meet PLOS ONE’s publication criteria as it currently stands. Therefore, we invite you to submit a revised version of the manuscript that addresses the points raised during the review process.

I would like to thank the authors for a comprehensive effort to improve the manuscript after the first revision. Also, I would like to apologise for the delay, which was caused by difficulties with finding a new reviewer and also, additionally, I decided to review the statistical part of the writing myself.

I have two reviews on my desk now, but unfortunately with different recommendations: accepted with minor comments (newly invited reviewer3), and a major revision with more serious remarks (reviever2); some of which I will bring up below:

1) Statistical analyses were not performed rigorously. In this case, I reviewed the statistical analyses by myself and the comments could be found below in the 'Editorial Statistical revision' section.

2) The results do not support the conclusions. The stress levels were confused by the authors with temperature levels. Studied populations were not directly comparable because there were different stress levels on each (the Arctic 10-20°C increase vs Helgoland 2-7°C increase). Therefore, the conclusion about a higher stress tolerance in Helgoland population is not supported and could be the result of different experimental design.

3) The experimental design and results did not allow to conclude about evolutionary consequences related to adaptation, as the authors suggested. The results should be interpreted more directly and as straight as possible. Moreover, the response for acclimation/acclimatisation should not be confused with adaptation.

The reviewer2 and my major comments are related to the main thesis and their complementarity, therefore my decision for this manuscript is Major revision.

We would appreciate receiving your revised manuscript by Feb 24 2020 11:59PM. To enhance the reproducibility of your results, we recommend that if applicable you deposit your laboratory protocols in protocols.io, where a protocol can be assigned its own identifier (DOI) such that it can be cited independently in the future. For instructions see: http://journals.plos.org/plosone/s/submission-guidelines#loc-laboratory-protocols

We look forward to receiving your revised manuscript.

Kind regards,

Adrian Zwolicki, Ph.D

Academic Editor

PLOS ONE

Additional Editor Comments (if provided):

Editorial Statistical revision

1) This study was performed on the insufficient number of replications. It is a statistically inappropriate idea to have more factor levels than samples per level (n=4). Such sample size is also insufficient to trust S-W normality test and, what is more important, it reduces the statistical power of ANOVA (likelihood of rejecting the H0 when it actually should be rejected, a type II error).

2) For the reason of heterogeneity of variance in factor levels and both types of distribution skewness in the data, the Box-Cox transformation should be performed prior to all analysis.

3) Because of low number of replications, PERMANOVA with PERMPDISP should be applied instead of all ANOVA type analyses, which would also help to unify the interpretation of the results. If the PERMDISP reviled heterogeneity, a short information (*) in the tables is required and short comments in results or discussion about how data quality could influence the results.

4) Despite the first reviewer’s suggestion, my recommendation is to use the same level of alpha α = 0.05 in all models/tests to unify their sensitivity, and to allow for comparisons between the models and results interpretation. The significance level should be mentioned only once in the methods section and removed from the results and tables.

5) In all PERMANOVA tables the SS for residuals (unexplained variation) should also be shown. It allows the readers to calculate the percent of total variation explained by tested factors if needed.

6) In all tables the exact value of probability should be presented with the exception of p<0.001. The values of SS, MS and pseudo-F should be rounded to two decimal places.

7) The “Statistical analysis for” phrase should be removed from all tables’ captions and the name of analysis - PERMANOVA should be added. Also “Post-hoc analyses were performed using pair-wise t-test comparisons” should be changed to “The post-hoc results are presented in Fig. X’.

Reviewers' comments:

Reviewer's Responses to Questions

**Comments to the Author**

1. If the authors have adequately addressed your comments raised in a previous round of review and you feel that this manuscript is now acceptable for publication, you may indicate that here to bypass the “Comments to the Author” section, enter your conflict of interest statement in the “Confidential to Editor” section, and submit your "Accept" recommendation.

Reviewer #2: (No Response)

Reviewer #3: (No Response)

2. Is the manuscript technically sound, and do the data support the conclusions?

Reviewer #2: No

Reviewer #3: Yes

3. Has the statistical analysis been performed appropriately and rigorously? 

Reviewer #2: No

Reviewer #3: Yes

4. Have the authors made all data underlying the findings in their manuscript fully available?

Reviewer #2: Yes

Reviewer #3: Yes

5. Is the manuscript presented in an intelligible fashion and written in standard English?

Reviewer #2: Yes

Reviewer #3: Yes

6. Review Comments to the Author

Reviewer #2: The authors downplayed the experimental concerns that the common garden experiment exposed the Arctic population to higher temperature stress (10-20°C increase) compared to the Helgolandic population (2-7°C increase), which has an implication on all response variables measured and consequently on the interpretation of results, (indirectly) suggesting that Helgoland population is more tolerant and Arctic population more susceptible to ocean warming.

The authors rebutted that “This paper does not aim to investigate whether populations will survive x degrees above their summer temperature. It aims to investigate whether this species has distinct thermal tolerances depending on the population of that same species; i.e., whether distinct populations of the same species have any functional differences at any temperatures. It is not an ecological question, it is an evolutionary question.”

The fact is they are comparing the two populations. How could they suggest one population is better than the other in tolerating stressful temperature condition, when absolute temperature values were used i.e. biased towards the habit temperature of one population and disregarded the ambient maximum summer temperatures the different populations are exposed to? The experiment could have designed +2, +4, +8, +16 °C increase relative to their highest summer temperature.

Organismal and/or population responses to ecological stress factors have evolutionary consequences. It seems inappropriate to suggest this study is answering evolutionary question but not ecological question. How does the experimental design answer which evolutionary question? The above contention needs to be contextualize in the paper.

The authors are advised to reassess or tone down their data interpretation and discussion relative to the limitation of their experimental design.

Reviewer #3: This study examines the effects of different warming treatments on survival and performance of early life stages of kelp species from northern and southern range edges. The work is comprehensive and the results highly interesting, especially the larger conclusion that there are differences in thermal tolerances of geographically different populations of a species. Modeller often use a single thermal tolerance to predict range expansions or retractions of species with climate change, and results such as these challenge these simplisitic approaches and are highly important. The authors have done a good job of addressing the reviewers’ comments. However, I suggest that the temperature treatments are heat spikes, which is consistent with the Hobday et al. definition of extreme temperature events shorter than 5 days. I would also mention heatwaves in the ms again (e.g. revert to previous wording on line 62). The removal of this from the manuscript is a missed opportunity to tie these experiments in with the larger literature on MHWs. The authors have responded well to the criticisms of reviewer 2, and their further clarification of the research question was appropriate.

Minor comments by line.

Line 24. Arctic.

Line 128 5 sporophytes is not that many. Can you justify why so few adults are representative of the larger populations in these areas?

Line 266. didn’t should not be contracted here and throughout.

Table 1. report error

Line 382. For clarity, be more specific as to how they influence recruitment capacity.

Line 385. Similar to what? Would be helpful to be more specific.

Table 4. Check significant digits, there is no need for 3 decimal places.

Line 448. 28 °C conditions are survived for…

Line 452. Not sure you need (=L. schinzii) here, the species name pallida is established no?

Line 470. Why are the species names reported in this way… ‘L. schinzii (=L. pallida) [30]; Eckloniopsis’ ? This is not consistent with previous paragraph.

7. PLOS authors have the option to publish the peer review history of their article (what does this mean?). If published, this will include your full peer review and any attached files.

Reviewer #2: No

Reviewer #3: No

---

## [Author Response · Author response to Decision Letter 1]

23 May 2020

I have two reviews on my desk now, but unfortunately with different recommendations: accepted with minor comments (newly invited reviewer3), and a major revision with more serious remarks (reviever2); some of which I will bring up below:

1) Statistical analyses were not performed rigorously. In this case, I reviewed the statistical analyses by myself and the comments could be found below in the 'Editorial Statistical revision' section.

We very much appreciate the Editor’s suggestions with regard to the statistical analyses. All the recommended changes have now been implemented to improve the statistical analyses.

2) The results do not support the conclusions. The stress levels were confused by the authors with temperature levels. Studied populations were not directly comparable because there were different stress levels on each (the Arctic 10-20°C increase vs Helgoland 2-7°C increase). Therefore, the conclusion about a higher stress tolerance in Helgoland population is not supported and could be the result of different experimental design.

This point is addressed directly in our responses to reviewer #2

3) The experimental design and results did not allow to conclude about evolutionary consequences related to adaptation, as the authors suggested. The results should be interpreted more directly and as straight as possible. Moreover, the response for acclimation/acclimatisation should not be confused with adaptation.

This point will be addressed directly in our responses to reviewer #2. 

Editorial Statistical revision

1) This study was performed on the insufficient number of replications. It is a statistically inappropriate idea to have more factor levels than samples per level (n=4). Such sample size is also insufficient to trust S-W normality test and, what is more important, it reduces the statistical power of ANOVA (likelihood of rejecting the H0 when it actually should be rejected, a type II error).

We accept the criticism regarding the insufficient number of replications used in this study and therefore we implemented all the changes recommended by the Editor to improve the statistical power of the analyses (please see the answers below).

2) For the reason of heterogeneity of variance in factor levels and both types of distribution skewness in the data, the Box-Cox transformation should be performed prior to all analysis.

Only the normalized sporophyte density showed heterogeneity of variance and distribution skewness and therefore these data were transformed via Box-Cox as suggested. 

3) Because of low number of replications, PERMANOVA with PERMPDISP should be applied instead of all ANOVA type analyses, which would also help to unify the interpretation of the results. If the PERMDISP reviled heterogeneity, a short information (*) in the tables is required and short comments in results or discussion about how data quality could influence the results.

We appreciate the Editor’s suggestion and all data were analysed with PERMANOVA and PERMDISP to unify the results. Whenever PERMDISP revealed heterogeneity in significant main effects or interactions found with PERMANOVA, this information was included in the manuscript.

4) Despite the first reviewer’s suggestion, my recommendation is to use the same level of alpha α = 0.05 in all models/tests to unify their sensitivity, and to allow for comparisons between the models and results interpretation. The significance level should be mentioned only once in the methods section and removed from the results and tables.

The Editor’s suggestion was accepted. The level of alpha α = 0.05 was implemented in all tests. This information was included only in the methods section (Line 312).

5) In all PERMANOVA tables the SS for residuals (unexplained variation) should also be shown. It allows the readers to calculate the percent of total variation explained by tested factors if needed.

As suggested, the values of SS for residuals were included in all PERMANOVA tables. 

6) In all tables the exact value of probability should be presented with the exception of p<0.001. The values of SS, MS and pseudo-F should be rounded to two decimal places.

As recommended, the exact value of probability was shown in the tables and the SS, MS and pseudo-F values were decreased to two decimal places.

7) The “Statistical analysis for” phrase should be removed from all tables’ captions and the name of analysis - PERMANOVA should be added. Also “Post-hoc analyses were performed using pair-wise t-test comparisons” should be changed to “The post-hoc results are presented in Fig. X’.

The suggestion of the Editor was accepted and we have replaced “statistical analysis” by “PERMANOVA” and “Post-hoc analyses were performed using pair-wise t-test comparisons” by “The post-hoc results are presented in Fig. X” in all tables.

Reviewer #2: 

The authors downplayed the experimental concerns that the common garden experiment exposed the Arctic population to higher temperature stress (10-20°C increase) compared to the Helgolandic population (2-7°C increase), which has an implication on all response variables measured and consequently on the interpretation of results, (indirectly) suggesting that Helgoland population is more tolerant and Arctic population more susceptible to ocean warming.

The comments of the reviewer refer to a different research question than the one in this paper. This paper aims to discover if different populations of the same species have distinct thermal tolerance limits and responses. This paper never did aim to test local responses to future effects of global warming (by comparing increases in temperatures of the same number of degrees above the temperature in which populations live locally). The referee’s comments are very interesting for any other paper with a research question about effects of oceanic warming among distinct populations, but are not relevant to this paper which asks the research question of whether individuals of the same species have distinct thermal tolerance range limits, as hypothesized if there is local selection in populations living in distinct habitats. 

The authors rebutted that “This paper does not aim to investigate whether populations will survive x degrees above their summer temperature. It aims to investigate whether this species has distinct thermal tolerances depending on the population of that same species; i.e., whether distinct populations of the same species have any functional differences at any temperatures. It is not an ecological question, it is an evolutionary question.”

The fact is they are comparing the two populations. How could they suggest one population is better than the other in tolerating stressful temperature condition, when absolute temperature values were used i.e. biased towards the habit temperature of one population and disregarded the ambient maximum summer temperatures the different populations are exposed to? The experiment could have designed +2, +4, +8, +16 °C increase relative to their highest summer temperature.

Organismal and/or population responses to ecological stress factors have evolutionary consequences. It seems inappropriate to suggest this study is answering evolutionary question but not ecological question. How does the experimental design answer which evolutionary question? The above contention needs to be contextualize in the paper.

The authors are advised to reassess or tone down their data interpretation and discussion relative to the limitation of their experimental design.

By comparing populations near the cold and warm edges of the thermal envelope for this species, the goal is rather to test whether phenotypic traits in response to a common thermal stressor are stable over the species range, or whether they differ between populations from distinct environments, suggesting an underlying genetic component, that is an adaptive change between populations. In order to do this, a common garden approach was taken to remove as far as possible the effects of environmental history, hardening or pre-acclimation, so that they were exposed to the experimental conditions after living in a common thermal habitat for 3 years in the exact same temperatures. This is a standard approach to detect adaptation rather than acclimation. A 3-years common garden culture of haploid meiotic products (i.e., gametophytes) derived from the collected material, means that the cells under the experimental treatments had been dividing and growing in the exact same temperatures despite coming from distinct populations, as necessary to distinguish adaptation from acclimation. This is the best design for the stated null hypothesis that thermal phenotype is stable across the species thermal envelope, versus the alternative hypothesis that it differs between populations and that difference remains even when they have been living in the exact same thermal conditions. This design, by exposing the experimental material to a long-term common acclimation period followed by a common experimental design, therefore allows the inference of a genetic basis, that is adaptation and not acclimation, linked to past evolutionary history and selective forces, acting on the source population.

The reviewer suggests a design for an entirely different research question than ours outlined above, namely to assess whether temperature changes of a common delta magnitude relative to those prevailing at the population site would have a similar response magnitude and thereby similar or different ecological effects on each population. Although this is a meaningful research question, we did not follow this idea as we already possess a series of published and unpublished pre-information (Bolton and Lüning 1982, tom Dieck 1992, 1993, Bartsch et al. 2013, Martins et al. 2017, Franke 2019, King et al. 2019, Liesner et al. under revision) that suggest subtle response differences between populations of Laminaria digitata with respect to their thermal performance. Thus, the proposed delta treatments possibly would have generated quite foreseeable responses with the Helgoland individuals losing their fertility at sub-lethal temperatures and dying when surpassing their lethal limits and the Spitzbergen individuals entering their near optimal conditions. With our design and research question, in contrast, we are able to better resolve one aspect of the general response width of the meta-population of the investigated species which are e.g. required by modellers (see remark rev 3). We added a few words to the introduction (Line 91) highlighting the differences between these two research questions, ours more from a mechanistic point of view and the delta treatments concept better showing the direct ecological impacts under global warming, which are both relevant. We thereby hope to better inform the reader how to discriminate these two research cases that may not have been clear enough in our last manuscript version. 

Finally, an interesting remark is that such experiments have both evolutionary implications (with regard to functional diversity within the species) and ecological implications. This is because unique thermal phenotypes or variants near the warm edge, if lost due to future climate impacts, either from extreme events or general poleward range shifts, may not be easily replaceable by natural migration or management actions from more northern populations.

Bartsch I, Vogt J, Pehlke C, Hanelt D. Prevailing sea surface temperatures inhibit summer reproduction of the kelp Laminaria digitata at Helgoland (North Sea). J Phycol. 2013; 49: 1061-1073.

Bolton J, Lüning K. Optimal growth and maximal survival temperature of Atlantic Laminaria species (Phaeophyta) in culture. Mar Biol. 1982; 66: 89-94.

Franke, K. (2019). Performance of different life cycle stages of the Arctic kelps Laminaria digitata and Saccharina nigripes along temperature gradients. Master's thesis. Bremen, Germany: University of Bremen.

King NG, McKeown NJ, Smale DA, Wilcockson DC, Hoelters L, Groves EA, et al. Evidence for different thermal ecotypes in range centre and trailing edge kelp populations. J Exp Mar Bio Ecol. 2019; 514-515: 10-17.

Liesner D, Fouqueau L, Valero M, Roleda MY, Pearson GA, Bischof K, Valentin K, Bartsch I. Heat stress responses and population genetics of the kelp Laminaria digitata (Phaeophyceae) across latitudes reveal differentiation among North Atlantic populations. Submitted

Martins N, Tanttu H, Pearson GA, Serrão EA, Bartsch I. Interactions of daylength, temperature and nutrients affect thresholds for life stage transitions in the kelp Laminaria digitata (Phaeophyceae). Bot Mar. 2017; 60: 109-121.

tom Dieck I. North Pacific and North Atlantic digitate Laminaria species (Phaeophyta): hybridization experiments and temperature responses. Phycologia. 1992; 31: 147-163.

tom Dieck I, de Oliveira EC. The section Digitatae of the genus Laminaria (Phaeophyta) in the northern and southern Atlantic: crossing experiments and temperature responses. Mar Biol. 1993; 115: 151-160.

Reviewer #3: 

This study examines the effects of different warming treatments on survival and performance of early life stages of kelp species from northern and southern range edges. The work is comprehensive and the results highly interesting, especially the larger conclusion that there are differences in thermal tolerances of geographically different populations of a species. Modeller often use a single thermal tolerance to predict range expansions or retractions of species with climate change, and results such as these challenge these simplisitic approaches and are highly important. The authors have done a good job of addressing the reviewers’ comments. However, I suggest that the temperature treatments are heat spikes, which is consistent with the Hobday et al. definition of extreme temperature events shorter than 5 days. I would also mention heatwaves in the ms again (e.g. revert to previous wording on line 62). The removal of this from the manuscript is a missed opportunity to tie these experiments in with the larger literature on MHWs. 

Because our research question is only and simply about discovering whether populations acclimated to common thermal conditions still remain different in thermal responses, at a particular set of temperatures in which such differences could be expected to happen, it is not really important for the question whether the tested set is called a heat wave or not. We are happy to leave the description of the experimental temperatures as it is, or to call it a heat wave. It really does not affect the research question or the conclusions, so we leave it up to the editor to decide whether to use this designation or not, given the disagreement among reviewers.

Minor comments by line.

• Line 24. Arctic.

As recommended, Artic was corrected to Arctic. 

• Line 128 5 sporophytes is not that many. Can you justify why so few adults are representative of the larger populations in these areas?

We would like to clarify that we did not use sporophytes but rather male and female gametophytic progeny of each sporophyte. We agree that it would be preferable to use a wider gene pool from more parental sporophytes for the comparison of populations, but establishment of a next generation of individual gametophyte cultures is very labour- and time-consuming and this was the maximum manageable number of replicates. To generate sufficient vegetative gametophyte material to perform a replicated experiment such as ours requires at least 1-1.5 years (Bartsch 2018). In addition, we need the cultivation space. Thus to isolate and culture a sub-set of individual isolates from 5 sporophytes represents a good cost-time-space balance. The advantage of using individual gametophyte cultures is to investigate isolates from different populations/sites at the same time and cultivated under common conditions since the meiospore stage, thereby allowing to distinguish adaptation from acclimation/environmental history.

• Line 266. didn’t should not be contracted here and throughout.

In accordance with the reviewer comment, didn’t was changed to did not throughout the manuscript.

• Table 1. report error

As suggested, the values for residuals or errors were included in all statistical tables.

• Line 382. For clarity, be more specific as to how they influence recruitment capacity.

As recommended, this information was included in the text (Line 439).

• Line 385. Similar to what? Would be helpful to be more specific.

Similar recruitment patterns were observed in the normalized sporophyte density (Fig. 5) and the relative presence of female multicellular gametophytes with sporophytes (S2 Fig). As suggested, this information was included in the manuscript (Line 444).

• Table 4. Check significant digits, there is no need for 3 decimal places.

As recommended, the decimal digits were decreased in all statistical tables.

• Line 448. 28 °C conditions are survived for…

The gametophytes of L. digitata survived at 28ºC for 1 day as already described in the text. However, we recognize that this sentence was not clear and, therefore, we clarified this in the discussion section (Line 516). 

• Line 452. Not sure you need (=L. schinzii) here, the species name pallida is established no?

Indeed, the species name L. pallida is established; however the study that we cited in the text was performed by tom Dieck in 1993, when it was still known as L. schinzii. Therefore, we decided to keep both the current and previous species names, so that the readers that are not familiar with the species name change can associate the tom Dieck (1993) work by realizing that L. schinzii was in fact L. pallida.

• Line 470. Why are the species names reported in this way… ‘L. schinzii (=L. pallida) [30]; Eckloniopsis’ ? This is not consistent with previous paragraph.

We agree with the Reviewer comment and we reported the current and previous species names in a consistent way thorough the manuscript, i.e., current species name [formerly known as]. L. pallida [=L. schinzii)] and Ecklonia radicosa [=Eckloniopsis radicosa] (Line 538).

---

## [Decision Letter · Decision Letter 2]

9 Jun 2020

PONE-D-19-17288R2

Thermal traits for reproduction and recruitment differ between Arctic and Atlantic kelp Laminaria digitata

PLOS ONE

Dear Dr. Martins,

Thank you for submitting your manuscript to PLOS ONE. After careful consideration, we feel that it has merit but does not fully meet PLOS ONE’s publication criteria as it currently stands. Therefore, we invite you to submit a revised version of the manuscript that addresses the points raised during the review process.

One final minor remark, please adjust the length of the abstract to the requirements of the journal. Which allow me to finally accept the mauscript.

We look forward to receiving your revised manuscript.

Kind regards,

Adrian Zwolicki, Ph.D

Academic Editor

PLOS ONE

Reviewers' comments:

Reviewer's Responses to Questions

**Comments to the Author**

1. If the authors have adequately addressed your comments raised in a previous round of review and you feel that this manuscript is now acceptable for publication, you may indicate that here to bypass the “Comments to the Author” section, enter your conflict of interest statement in the “Confidential to Editor” section, and submit your "Accept" recommendation.

Reviewer #2: All comments have been addressed

2. Is the manuscript technically sound, and do the data support the conclusions?

Reviewer #2: Yes

3. Has the statistical analysis been performed appropriately and rigorously? 

Reviewer #2: Yes

4. Have the authors made all data underlying the findings in their manuscript fully available?

Reviewer #2: Yes

5. Is the manuscript presented in an intelligible fashion and written in standard English?

Reviewer #2: Yes

6. Review Comments to the Author

Reviewer #2: Authors adequately addressed the concerns and supplied information in the introduction. Abstract is too long. Please edit concisely.

7. PLOS authors have the option to publish the peer review history of their article (what does this mean?). If published, this will include your full peer review and any attached files.

Reviewer #2: No

---

## [Author Response · Author response to Decision Letter 2]

13 Jun 2020

Answers to the Academic Editor and Reviewer comments:

Academic Editor:

One final minor remark, please adjust the length of the abstract to the requirements of the journal. Which allow me to finally accept the manuscript.

Reviewer 2: 

Authors adequately addressed the concerns and supplied information in the introduction. Abstract is too long. Please edit concisely.

As recommended, the abstract was shortened to less than 300 words to meet the publication requirements of PLOS ONE.

---

## [Editor Report · Decision Letter 3]

16 Jun 2020

Thermal traits for reproduction and recruitment differ between Arctic and Atlantic kelp Laminaria digitata

PONE-D-19-17288R3

Dear Dr. Martins,

We’re pleased to inform you that your manuscript has been judged scientifically suitable for publication and will be formally accepted for publication once it meets all outstanding technical requirements.

Kind regards,

Adrian Zwolicki, Ph.D

Academic Editor

PLOS ONE
---

## [Editor Report · Acceptance letter]

19 Jun 2020

PONE-D-19-17288R3 

Thermal traits for reproduction and recruitment differ between Arctic and Atlantic kelp *Laminaria digitata*

Dear Dr. Martins:

I'm pleased to inform you that your manuscript has been deemed suitable for publication in PLOS ONE. Congratulations! Your manuscript is now with our production department. 

Kind regards, 

on behalf of

Dr. Adrian Zwolicki 

Academic Editor

PLOS ONE